# Understanding Particulate Matter Retention and Wash-Off during Rainfall in Relation to Leaf Traits of Urban Forest Tree Species

**Myeong Ja Kwak** [1], **Jongkyu Lee** [1], **Sanghee Park** [1], **Yea Ji Lim** [1], **Handong Kim** [1], **Su Gyeong Jeong** [1], **Joung-a Son** [2], **Sun Mi Je** [2], **Hanna Chang** [2], **Chang-Young Oh** [2], **Kyongha Kim** [1] and **Su Young Woo** [1,*]

1    Department of Environmental Horticulture, University of Seoul, Seoul 02504, Republic of Korea
2    Urban Forests Research Center, National Institute of Forest Science, Seoul 02455, Republic of Korea
*    Correspondence: wsy@uos.ac.kr; Tel.: +82-10-3802-5242

**Abstract:** Dynamic particulate matter (PM) behavior on leaves depends on rainfall events, leaf structural and physical properties, and individual tree crowns in urban forests. To address this dependency, we compared the observed relationships between PM wash-off ability and leaf traits on inner and outer crown-positioned leaves during rainfall events. Data showed significant differences in the PM wash-off ability between inner and outer crown-positioned leaves relative to rainfall events due to leaf macro- and micro-structure and geometric properties among tree species. Our results showed that PM wash-off effects on leaf surfaces were negatively associated with trichome density and size of leaf micro-scale during rainfall events. Specifically, *Quercus acutissima* with dense trichomes and micro-level surface roughness with narrow grooves on leaf surfaces showed lower total PM wash-off in both inner (−38%) and outer (105%) crowns during rainfall. Thus, their rough leaves in the inner crown might newly capture and/or retain more PM than smooth leaves even under rainfall conditions. More importantly, *Euonymus japonicus*, with a thin film-like wax coverage without trichome, led to higher total PM wash-off in both inner (368%) and outer (629%) crowns during rainfall. Furthermore, we studied the changes in PM wash-off during rainfall events by comparing particle size fractions, revealing a very significant association with macro-scale, micro-scale, and geometric features.

**Keywords:** inner and outer crown; leaf macro- and micro-structure; particulate matter; PM wash-off; rainfall

## 1. Introduction

Urban vegetation comprises individual trees that contribute to sustainable ecosystems. Due to their spatial and seasonal biomass distribution, individual trees can regulate local climatic conditions by reducing gaseous and particulate pollutants [1] and by modifying the abundance and distribution of environmental elements such as water and light [2]. As part of the biosphere, trees interact with water and the atmosphere over different scales [2]; land use, weather condition, tree canopy, crown shape, leaf spatial arrangement, and leaf ultrastructure can lead to variations in particulate matter (PM) deposited on leaf surfaces [3]. The potential PM generation has increased because of urbanization, industrialization, and developmental expansion, which has raised global concerns and may negatively influence human health and the environment [4,5].

Urban trees can be widely used as biological filters that intercept airborne PM (e.g., *Buxus koreana* [6], *Cedrus deodara* [5], *Euonymus japonicus* [6–8], *Pinus tabulaeformis* [5,7], *Sophora japonica* [7,8], *Taxus cuspidata* [6], and *Ulmus pumila* [5,8]). Previous studies have demonstrated that PM-capturing and -retaining capacities on leaves depend on surface roughness and microstructure properties [5–10]. Among various weather factors, a better understanding of the underlying physical processes between wash-off and redeposition of

PM particles on leaf surfaces by rainfall events is important for air quality dynamics [10,11]. PM particles adsorbed on leaf surfaces are resuspended into the atmosphere by wind. Net removal of PM particles from the leaf surfaces to the ground can be achieved through a wash-off process during rainfall [9,12,13]. Therefore, the wash-off process by rainfall can be considered a key factor for restoring the barrier function to filter the PM on plant leaves [7,13]. Plant species growing in polluted areas may undergo morphological changes such as stomata, trichomes, surfaces, and cuticles, affecting overall photosynthesis, stomatal conductance, and transpiration rate. Thus, plants may considerably reduce their essential function as biological filters [14].

Other factors that may affect PM removal by rainfall are leaf shapes and surface traits (e.g., smooth surfaces [13], trichome [13], epicuticular wax [13,15], contact angle [15], and groove [15]) when raindrops hit leaf surfaces, the hydrological characteristics of rainfall events, and PM retention mass before the rainfall events [9,12,13,15]. The dry and wet deposition processes are involved in PM removal as an important means of controlling air pollutants from the atmosphere to land surfaces [16–18]. Differences in leaf surface microstructures (e.g., cuticle and trichomes) influence water repellency and surface roughness [19]. These variations result in the leaf surfaces having different wetting properties, which is a major factor that increases or hinders PM removal [7,19,20]. The masses and rates of PM wash-off that vary by tree species are recognized as important key indicators of tree species selection for a barrier or filtration effect [15]. Furthermore, it is necessary to consider both retention and wash-off of PM in the inner and outer crown positions after rainfall events when screening for PM purification trees.

In many contexts, field experiments are subject to many limitations due to other environmental interference factors such as atmospheric humidity, temperature, and wind speed; thus, the rainfall-induced wash-off processes have been proactively carried out using rainfall simulation experiments [7,17,21,22]. Most field experiments have been conducted using deposition models, such as PM and canopy interception modeling at tree canopy levels, and fewer studies on PM wash-off efficiency in the inner and outer crowns of individual trees due to rainfall have been conducted [12]. Given the interest in direct field measurements, few studies have studied PM wash-off efficiency in individual trees' inner and outer crowns during rainfall intensity. On the other hand, extending the in-chamber seedling experiment to large trees growing in cities is not easy; hence, additional field research is required to explore the leaf potential in dynamic PM retention and wash-off processes in tree crowns [8,20]. It is necessary to recognize the barrier functions of urban forests to PM particles resuspended from urban lands containing different pollutants and to summarize information regarding the complex phenomenon of PM retention, wash-off, and resuspension under rainfall events. Here, we hypothesized that rainfall intensity could change PM retention and wash-off efficiencies depending on the inner and outer canopy location of urban tree species. These changes may depend on the unique microstructure and tree-specific crowns. Therefore, we described (1) the PM retention and wash-off in inner and outer crown-positioned leaves under natural rainfall intensity and (2) the relationship between PM retention and wash-off processes and leaf micro-structural factors.

## 2. Materials and Methods

### 2.1. Experimental Site and Plant Materials

The experimental site is the Seoul Forest Park, a public open green space located in Seongdong-gu which is famous as a transportation and commercial center in the middle of the metropolis in eastern Seoul; however, this site is close to the boundary of a ready-mix concrete plant to the west. There were six rounds of field measurements during different natural rainfall events in the Seoul Forest Park [6], targeting five representative species: Korean red pine (*Pinus densiflora* Siebold & Zucc.), sawtooth oak (*Quercus acutissima* Carruth.), dawn redwood (*Metasequoia glyptostroboides* Hu & W. C. Cheng), evergreen spindletree (*Euonymus japonicus* Thunb.), and Korean flowering cherry (*Prunus yedoensis* Matsum.). We collected $n = 120$ twigs from branches in four quadrant directions in inner and outer crowns at a tree height of 3 to

5 m from a total of $n = 5$ individual trees of each species at the Seoul Forest Park. Prior to transferring into the laboratory, the sampled twigs were placed in a paper bag and stored in an icebox. *M. glyptostroboides* is a deciduous conifer of the family Cupressaceae with a height of 35 m. The fern-like foliage is a pinnately compound leaf, opposite, and flat and generally appears two-ranked in a flat display. The average leaf area was 18.9 cm$^2$. *P. yedoensis* is a deciduous broad-leaved tree species of the Rosaceae family, with a height of 15 m. The leaves are alternate phyllotaxis, elliptical-ovate, have doubly serrate margins, and are 6–12 cm long, with a leaf area of 45.9 cm$^2$. *Q. acutissima* is a deciduous broad-leaved tree of the family Fagaceae, with a height of 20–25 m. The leaves are lanceolate to oblong, displaying a sharply serrated margin with bristle-tipped teeth, 10–20 cm long, with a leaf area of 39.0 cm$^2$. *E. japonicus* is an evergreen broad-leaved species with a height of 3 m. The leaves are opposite phyllotaxis, leathery, lustrous, deep green, obovate to narrowly oval, and 3–7 cm long, with a leaf area of 22.0 cm$^2$. In particular, the leaf surfaces were shiny and showed slightly serrated margins. *P. densiflora* is an evergreen coniferous tree of the family Pinaceae, with a height of 15–45 m. The evergreen needles are dark green and up to 8–9 cm long, with two needles per fascicle. Generally, rainfall events can have different impacts depending on the amount of rainfall over a short period. Therefore, the degree of influence of rainfall intensity on PM wash-off may vary across the hourly rainfall magnitudes (i.e., short-term intensive rainfall) than the daily cumulative rainfall. According to the Korea Meteorological Administration data (KMA), rainfall intensity is generally classified into four categories: light (<3 mm/h), moderate (3–15 mm/h), heavy (15–30 mm/h), and violent (>30 mm/h), based on the rate of precipitation, which depends on the considered time. This study analyzed net PM removal ability change during natural rainfall events (19.5 mm heavy rain over three h) on surface PM and in-wax PM from the inner and outer crown-positioned leaves in five tree species. The meteorological-related information referred to the KMA. In addition, rainfall data were obtained from KMA Weather Data Service.

### 2.2. Gravimetric Determination of PM10 and PM2.5 Adsorbed on Leaf Surfaces and Waxes

Determination of PM particle size fractions on the collected leaves from the inner and outer individual tree crowns during the natural rainfall pattern was performed according to the modified method of Kwak et al. [6]. For each species, the sampled leaves (approximately 150–250 cm$^2$) were placed in individual glass beakers. Subsequently, the amount of PM adsorbed on leaf surfaces (SPM) and in wax (WPM) was measured gravimetrically using the ultrapure water washing method and the chloroform washing method, as shown in Table S1. The PM retention and wash-off effects on surfaces and in wax of inner and outer crown-positioned leaves during rainfall events were evaluated and compared before and after receiving rainfall at rainfall of 40 mm for 2 days. The PM wash-off capacities ($W_{\text{leaf}}$) in different particle size fractions before and after the rainfall event was calculated using the formula [23]:

$$W_{\text{leaf}} = \frac{\text{BR}_f - \text{AR}_f}{2}, \ \left[\frac{\text{mg/m}^2}{\text{d}}\right], \tag{1}$$

where $\text{BR}_f$ is the amount of each particle size fraction retained per leaf area before rainfall events, in mg/m$^2$; $\text{AR}_f$ is the amount of each particle size fraction retained per leaf area after rainfall events, in mg/m$^2$; and 2 is a two-day period of rainfall considered in this study, in d. Consequently, a positive value represents PM wash-off and otherwise a negative value indicates PM retention and/or newly captured PM particles. The particle size fractions were subdivided into fine particles (SPM2.5 and WPM2.5, diameter < 2.5 μm) and coarse particles (SPM10 and WPM10, diameter 2.5–10 μm) on leaf surfaces and in wax, respectively. Total PM (TPM) was assessed by summing the amounts of PM (i.e., SPM10 + WPM10) on the leaf surface and in wax.

### 2.3. Determination of Leaf Micro- and Macro-Morphological Features

Previous studies have shown that leaf micro- and macro-morphological features, such as roughness, stomata, trichomes, leaf margin complexity, leaf shape, width-to-length ratio

(W/L), and phenological type, are distinct driving factors for PM capture efficiency [8,21,24]. Furthermore, the values associated with PM capture capability can be linked to variable abundance/morphology or the presence/absence of specific characteristics [24]. Specifically, high scores have been associated with leaf surfaces with high percentages of roughness area [25], trichome-covered area [26], high stomata density [27], and small groove dimensions [5]. Trichome density, expressed as the percentage of trichomes on the leaf area, was measured using ImageJ software (version 1.53q) [24].

Data from the net PM wash-off ability obtained from the inner and outer crown-positioned leaves during rainfall events were used to evaluate associations with leaf macro-scale, micro-scale, and geometric properties, respectively. Macro-scale morphological variables included in our study were leaf area (LA), perimeter (P), circularity (C), leaf width-to-length ratio (W/L), vein-to-blade ratio (V/B), and leaf roundness index (RI). Micro-scale morphological variables included in our study were stomatal length (SLab) and width (SWab) on abaxial surfaces, stomatal density on adaxial (SDad) and abaxial (SDab) leaf surfaces, trichome density on vein (TVDad) and blade (TBDad) of adaxial surfaces, trichome density on vein (TVDab) and blade (TBDab) of abaxial surfaces, trichome length on vein (TVLad) and blade (TBLad) of adaxial surfaces, and trichome length on vein (TVLab) and blade (TBLab) of abaxial surfaces. Furthermore, geometric properties included in our study were leaf epicuticular wax (Wax), contact angle on adaxial (CAad) and abaxial (CAab) leaf surfaces, and micro-roughness on adaxial (RAad) and abaxial (RAab) leaf surfaces.

The micromorphological features of leaf surfaces were examined using field emission scanning electron microscopy (FESEM) equipped with energy dispersive X-ray spectroscopy (EDS), according to the method described by Kwak et al. [6]. To observe the microstructure of leaves, samples were collected from trees growing in the nursery of the National Institute of Forest Science. Five to ten fully developed leaves were collected per species, placed in Petri dishes, stored in an icebox, and transported to the laboratory. Thereafter, five leaves were placed on a leaf area measurement board and photographed, and the captured images were measured using WinFOLIA PRO 2013 software (WinFO-LIA, Regent Instruments, Québec, QC, Canada). After measuring the leaf area, the icebox containing the sample was transferred from the National Institute of Forest Science to the Urban Environment Laboratory at the University of Seoul, and the samples were pretreated for leaf surface observation. Three leaf fragments containing midribs and veins were cut into 5 mm × 20 mm pieces for broad-leaved trees and 20 mm for conifers, placed on a stub for electron microscopy with conductive double-sided carbon tape, and then dehydrated for 24 h using a vacuum freezing dryer to preserve the morphological features of the leaf surfaces (FD-8508, Ilshinbiobase, Co., Ltd., Dongducheon, Korea). The stub on which the freeze-dried leaf fragments were attached was stored in a desiccator chamber in which temperature and humidity were controlled until observation with an electron microscope. For electron microscopy, the samples were platinum-coated using an ion sputter coater (Ion-sputter, MC1000, Hitachi, Ibaraki, Japan) and imaged under a scanning electron microscope (SU8010, Hitachi High-Tech, Tokyo, Japan) at the Center for Research Facilities at the University of Seoul. The features were observed at three magnifications (×60, ×300, and ×2000 magnification) and then imaged. Subsequently, an optical microscope (Bruker, Contour GT-K) was used for 3D surface analysis of the sample for which the scanning electron microscope observation was completed. Leaf surface roughness was measured to analyze the differences among tree species.

### 2.4. Determination of Leaf Epicuticular Wax

Leaf discs for wax analysis were collected from leaves using a 0.8 mm-diameter disc punch. Twenty leaf punches of 0.8 cm diameter were collected and used to determine epicuticular wax concentrations using the colorimetric method. Leaf epicuticular wax was extracted by submerging leaf discs in 1 mL chloroform for 30 s, and the submersion time was previously determined to completely remove the epicuticular wax from the leaves. The resulting mixture was transferred to a clean 2 mL glass vial. The resulting

extract was oxidized by adding 300 μL acidified potassium dichromate and heated for 30 min in a heating block at 100 °C. After boiling, the vials were allowed to cool, and 700 μL of deionized water was added to each vial, allowing color to develop for 1 h before measurement. A spectrophotometer (PHERAstar plus, BMG Labtech, Offenburg, Germany) was used to determine the optical density of each sample at 590 nm. Carnauba wax was used to create a standard curve that was used to calculate wax levels based on leaf area.

### 2.5. Statistical Analysis

Data were analyzed to determine statistically significant differences using IBM SPSS Statistics 26 (SPSS Inc., IBM Company Headquarters, Chicago, IL, USA). We also performed Tukey's HSD multiple comparisons to identify the effects among species in net PM wash-off ability during rainfall for each PM particle size fraction at $p \leq 0.05$. We employed the non-parametric Kruskal–Wallis test to assess whether statistically significant structural differences of roughness and epicuticular wax exist among five tree species for PM retention and wash-off. The statistical significance of the difference in leaf roughness between adaxial and abaxial leaf surfaces of five species was tested using the non-parametric two-tailed Mann–Whitney U tests.

Specifically, the correlation of the net PM wash-off ability from the inner and outer crown-positioned leaves collected during rainfall events and leaf macro-scale, micro-scale, and geometric properties was performed via Pearson's correlation analysis. Then, correlograms were constructed via open-source software in RStudio using the "cor" built-in function and the publicly available package "corrplot" to illustrate the association between the pairs of variables.

## 3. Results

### 3.1. Variation in Leaf-Surface and In-Wax PM Mass in Leaf Samples Taken for Each Crown Position before and after Rainfall Events

During the rainfall event, there were significant differences among tree species in particle size fractions for both crown-positioned leaves, except WPM10 (Figure 1a) and WPM2.5 (Figure 1b). Of note, SPM10 was a noticeable difference among species in inner and outer crown-positioned leaves (Figure 1). Inner crown-positioned leaves showed significant differences among species for SPM2.5, SPM10, and WPM2.5 during the rainfall event (Figure 1a); outer crown-positioned leaves showed significant differences in SPM2.5, SPM10, and WPM10 (Figure 1b).

In the inner tree crown-positioned leaves (Figure 1a), SPM2.5 showed higher wash-off in *M. glyptostroboides* and *P. densiflora*, except for *P. yedoensis*, *E. japonicus*, and *Q. acutissima*. In addition, SPM10 showed obviously higher wash-off in *M. glyptostroboides* and *E. japonicus* and their order were *M. glyptostroboides* > *E. japonicus* > *P. yedoensis* > *P. densiflora* > *Q. acutissima*. WPM2.5 had a higher PM wash-off level in the following orders: *P. densiflora* > *Q. acutissima* > *M. glyptostroboides* > *P. yedoensis* > *E. japonicus*. Interestingly, WPM10 showed no statistically significant difference among species.

For outer tree crown-positioned leaves (Figure 1b), SPM2.5 showed a higher wash-off level in *P. densiflora*, in the following order: *P. densiflora* (8.116 mg m$^{-2}$ d$^{-1}$) > *M. glyptostroboides* (4.153 mg m$^{-2}$ d$^{-1}$) > *Q. acutissima* (1.317 mg m$^{-2}$ d$^{-1}$) > *E. japonicus* (0.841 mg m$^{-2}$ d$^{-1}$) > *P. yedoensis* (0.504 mg m$^{-2}$ d$^{-1}$). In addition, SPM10 showed relatively high wash-off levels in *E. japonicus* and *M. glyptostroboides*, in the following order: *E. japonicus* (33.158 mg m$^{-2}$ d$^{-1}$) > *M. glyptostroboides* (27.521 mg m$^{-2}$ d$^{-1}$) > *P. yedoensis* (12.733 mg m$^{-2}$ d$^{-1}$) > *P. densiflora* (11.915 mg m$^{-2}$ d$^{-1}$) > *Q. acutissima* (5.534 mg m$^{-2}$ d$^{-1}$). For PM particles encapsulated within wax, no significant differences in WPM2.5 levels were observed among species during the rainfall event, whereas WPM10 wash-off levels were relatively high, in the following order: *P. yedoensis* (3.185 mg m$^{-2}$ d$^{-1}$) > *P. densiflora* (2.418 mg m$^{-2}$ d$^{-1}$) > *M. glyptostroboides* (1.585 mg m$^{-2}$ d$^{-1}$) > *E. japonicus* (1.443 mg m$^{-2}$ d$^{-1}$) > *Q. acutissima* (0.227 mg m$^{-2}$ d$^{-1}$).

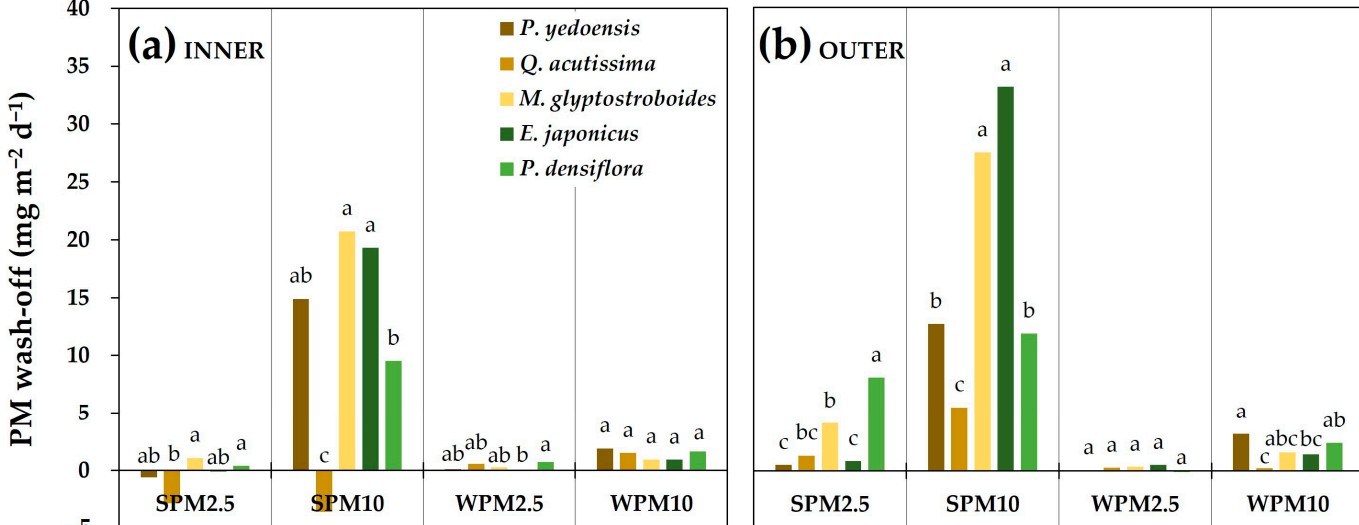

**Figure 1.** Net PM wash-off ability during rainfall events on surface PM (SPM2.5, SPM10) and in-wax PM (WPM2.5, WPM10) from the inner and outer crown-positioned leaves, based on the division into $PM_{2.5}$ and $PM_{10}$ fractions. (**a**) INNER, inner of individual tree crowns, (**b**) OUTER, outer of individual tree crowns. Different letters above bars represent significantly different among species in each PM particle size fraction at $p \leq 0.05$ using Tukey's HSD multiple comparisons. Data are mean, $n = 5$.

A clear difference is shown in PM wash-off levels between inner and outer crown-positioned leaves in each tree species during the rainfall event (Figure 2). Based on tree crown positions, SPM2.5 had higher wash-off levels in the outer tree crown-positioned leaves for *P. densiflora* (Figure 2e), *M. glyptostroboides* (Figure 2c), and *Q. acutissima* (Figure 2b), except for *P. yedoensis* (Figure 2a) and *E. japonicus* (Figure 2d). SPM10 had higher wash-off levels in the outer tree crown-positioned leaves for *E. japonicus*, *M. glyptostroboides*, and *Q. acutissima*, except for *P. yedoensis* and *P. densiflora*. *P. yedoensis* showed slightly higher wash-off of only WPM10 between the inner and outer crown-positioned leaves, whereas no significant difference was noted in SPM2.5, SPM10, WPM2.5, and TPM (Figure 2a). *Q. acutissima* showed higher PM retention and/or new PM capture of SPM2.5 and SPM10 in the inner tree crown-positioned leaves even during the rainfall event (Figure 2b). *M. glyptostroboides* had significant wash-off of SPM2.5, SPM10, WPM10, and TPM in the outer crown-positioned leaves during the rainfall event, except for WPM2.5 (Figure 2c). More importantly, *E. japonicus* showed obvious wash-off levels of WPM2.5 during the rainfall event in outer crown-positioned leaves compared to inner tree crown-positioned leaves (Figure 2d). Importantly, we found no statistically significant differences in SPM10, WPM2.5, WPM10, and TPM levels between inner and outer crown-positioned leaves of *P. densiflora* during a rainfall event, except for SPM2.5 (Figure 2e).

Based on the direct in-situ measurements of LAI (Figure 3), SPM2.5 showed a higher wash-off level in *P. densiflora*, in the following order: *P. densiflora* (7.075 mg $\mathrm{m}^{-2}$ $\mathrm{d}^{-1}$) > *M. glyptostroboides* (4.912 mg $\mathrm{m}^{-2}$ $\mathrm{d}^{-1}$) > *E. japonicus* (0.807 mg $\mathrm{m}^{-2}$ $\mathrm{d}^{-1}$). Importantly, it was quite interesting to find in our experiment that there were slightly higher PM particles on leaf surfaces (i.e., PM retention and/or new PM capture) in *Q. acutissima* and *P. yedoensis* during the rainfall event. In addition, SPM10 showed relatively high wash-off levels in *E. japonicus* and *M. glyptostroboides*, in the following order: *E. japonicus* (58.760 mg $\mathrm{m}^{-2}$ $\mathrm{d}^{-1}$) > *M. glyptostroboides* (45.44 mg $\mathrm{m}^{-2}$ $\mathrm{d}^{-1}$) > *P. yedoensis* (28.869 mg $\mathrm{m}^{-2}$ $\mathrm{d}^{-1}$) > *P. densiflora* (17.863 mg $\mathrm{m}^{-2}$ $\mathrm{d}^{-1}$) > *Q. acutissima* (2.047 mg $\mathrm{m}^{-2}$ $\mathrm{d}^{-1}$). For PM particles encapsulated within the wax, no significant differences in WPM2.5 levels were observed among species during the rainfall event, whereas WPM10 had relatively high wash-off levels in *P. yedoensis*, in the following order: *P. yedoensis* (5.311 mg $\mathrm{m}^{-2}$ $\mathrm{d}^{-1}$) > *P. densiflora* (3.375 mg $\mathrm{m}^{-2}$ $\mathrm{d}^{-1}$) > *E. japonicus* (2.64 mg $\mathrm{m}^{-2}$ $\mathrm{d}^{-1}$) > *M. glyptostroboides* (2.379 mg $\mathrm{m}^{-2}$ $\mathrm{d}^{-1}$) > *Q. acutissima* (1.809 mg $\mathrm{m}^{-2}$ $\mathrm{d}^{-1}$). TPM had relatively high wash-off levels in *E. japonicus*, in the following

order: *E. japonicus* (61.4 mg m$^{-2}$ d$^{-1}$) > *M. glyptostroboides* (47.819 mg m$^{-2}$ d$^{-1}$) > *P. yedoensis* (34.18 mg m$^{-2}$ d$^{-1}$) > *P. densiflora* (21.239 mg m$^{-2}$ d$^{-1}$) > *Q. acutissima* (3.856 mg m$^{-2}$ d$^{-1}$).

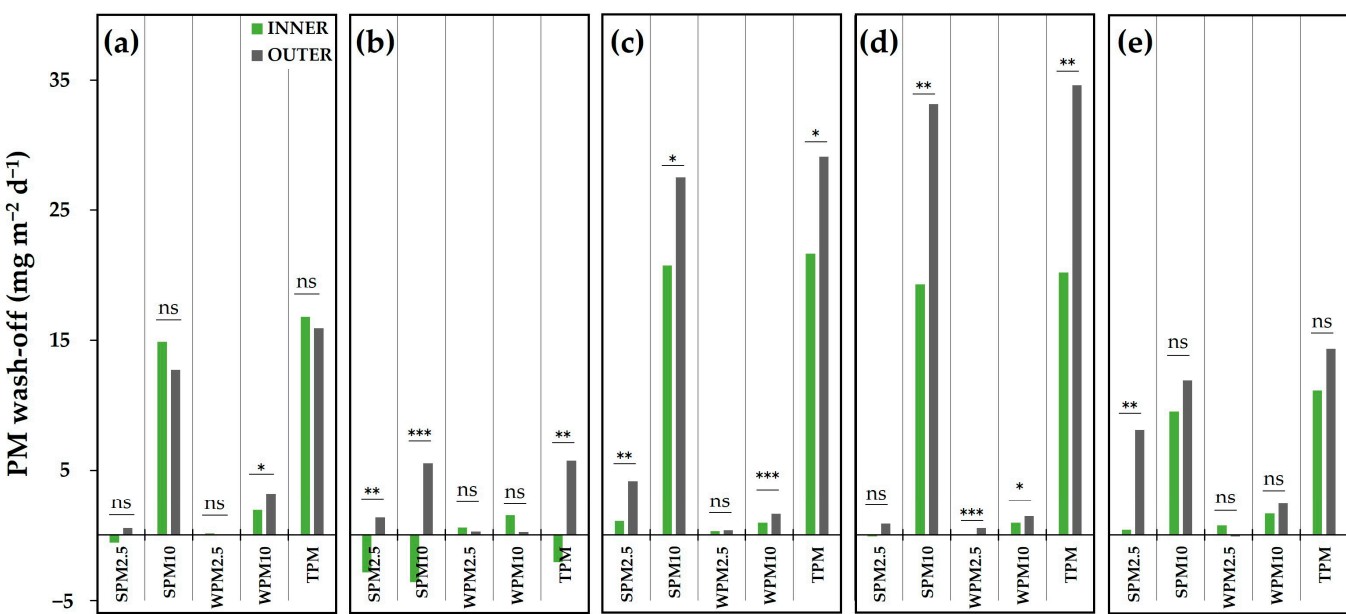

**Figure 2.** Net PM wash-off ability during rainfall events between inner (INNER) and outer (OUTER) crown-positioned leaves in each PM particle size fraction. (**a**) *P. yedoensis*, (**b**) *Q. acutissima*, (**c**) *M. glyptostroboides*, (**d**) *E. japonicus*, and (**e**) *P. densiflora*. Data are mean, *n* = 5. Bars with asterisks denote statistical differences between inner (INNER) and outer (OUTER) crown-positioned leaves in each PM particle size fraction within each tree species (* *p* < 0.05; ** *p* < 0.01; *** *p* < 0.001; ns: not significant, *p* > 0.05). Abbreviations: SPM2.5, surface PM2.5 on leaves; SPM10, surface PM10 on leaves; WPM2.5, in-wax PM2.5; WPM10, in-wax PM10. Note: negative numbers for PM retention.

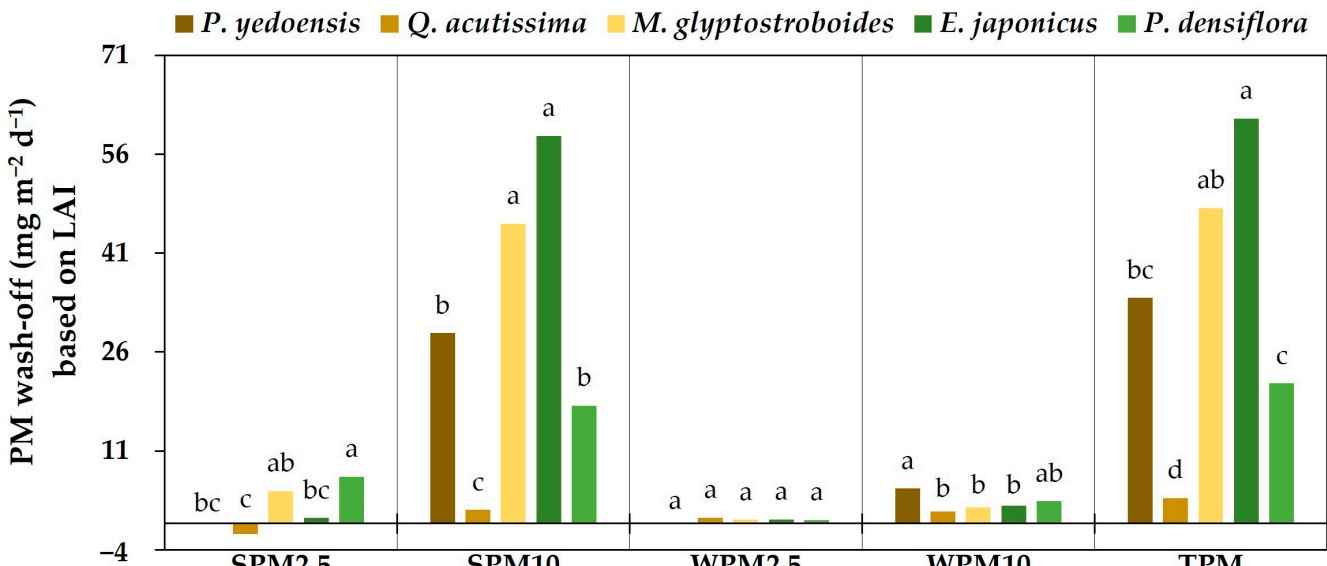

**Figure 3.** Net PM wash-off ability during rainfall events on surface PM (SPM2.5, SPM10), in-wax PM (WPM2.5, WPM10), and total PM (TPM) from the entire tree crown based on the LAI. Different letters above bars represent significant difference among species in each PM particle size fraction at *p* ≤ 0.05 using Tukey's HSD multiple comparisons. Data are mean, *n* = 10.

*3.2. Interspecific Trends in Variations of PM Adsorption in Surfaces and Wax Layers of Inner and Outer Crown Leaves before and after Rainfall Events*

As shown in Figure S1, positive and negative changes in leaf surface and in-wax PM particles after different rainfall intensities for five tree species were represented as categorical data and presented as percentages (i.e., PM retention (+) and wash-off (−) processes). There were no statistically significant changes in PM mass for *E. japonicus* ($t = -1.598$ and $p = 0.142$), *P. yedoensis* ($t = -1.835$ and $p = 0.083$), and *Q. acutissima* ($t = -0.878$ and $p = 0.391$) after light rainfall intensity. More importantly, PM retention effects were observed in *P. densiflora* ($t = 5.004$ and $p = 0.000$) and *M. glyptostroboides* ($t = 3.317$ and $p = 0.004$) after light rainfall intensity. Under moderate rainfall intensity, *P. densiflora* was found to have significant wash-off levels of −57% ($t = -5.613$ and $p = 0.000$) in PM mass, whereas *Q. acutissima* showed significantly higher PM mass ($t = 3.383$ and $p = 0.003$), clearly indicating that the retention of PM particles continually occurred throughout the rainfall record period. For heavy rainfall intensity, wash-off levels for both leaf surface and in-wax PM were found to be significantly higher in *E. japonicus* ($t = -7.429$ and $p = 0.000$), *M. glyptostroboides* ($t = -6.207$ and $p = 0.000$), *P. densiflora* ($t = -3.852$ and $p = 0.001$), and *P. yedoensis* ($t = -10.344$ and $p = 0.000$), except for *Q. acutissima* ($t = -0.173$ and $p = 0.865$).

*3.3. Determination of Leaf Microstructure and Surface Roughness*

The micromorphological, morphological, and 3D topography images of the leaf surfaces of each tree species are shown in Figures 4 and 5, respectively. For the studied tree species, the dominant surface structure was rough, indicating the existence of grooves, trichomes, stomata, and waxy cuticles (Figures 4–6). Grooves occurred on the adaxial surfaces of *M. glyptostroboides* and *P. yedoensis*, especially relatively striped deep grooves along the leaf vein for *M. glyptostroboides* (Figure 4a,b). *M. glyptostroboides* showed high roughness with deep grooves and densely distributed wax crystals on both leaf surfaces with no trichome (Figures 4a–c and 5a,b). In *P. yedoensis*, hairy trichomes were distributed on leaf blades or concentrated along veins (especially abaxial leaf surfaces), and leaf roughness was similar for both leaf surfaces (Figures 4d–f and 5c,d). Among the studied tree species, *Q. acutissima* showed non-glandular and stellate trichomes on leaf blades of abaxial leaf surfaces (Figures 4g–i and 5e,f) and had the lowest wax and surface roughness levels (Figure 5k,l). Meanwhile, *E. japonicus* showed relatively less rough and smoother surfaces without trichomes than other species. Instead, their abaxial surfaces displayed significantly higher surface roughness (Figure 4k), approximately three times that of adaxial surfaces (Figures 4j and 5k). *P. densiflora* had similar surface roughness on adaxial and abaxial needles and clearly showed higher cuticular wax amounts, similar to those found in *M. glyptostroboides* (Figures 4m,n and 5l). Specifically, *P. yedoensis* showed relatively high leaf surface roughness with coarse and dense grooves (Figures 5k and 6a). In contrast, *Q. acutissima* showed more obvious fine ridges on their adaxial surfaces (Figures 5e and 6b).

The calculated values of leaf macro-scale, micro-scale, and geometric properties based on Figures 4 and 5 are shown in Table S2–S4. The detailed leaf microstructural features of Figures 4 and 5 are shown below, as shown in Figures 6 and S2. *P. yedoensis* generated similar surface roughness between the abaxial and adaxial leaf surfaces (Figure 6a). The leaves were hypostomatic (i.e., stomata only occur on the abaxial surface) with stomatal lengths of 12.0 µm and stomatal density of 524.0 per mm$^2$. The trichomes revealed a high density of 45.9 per mm$^2$ on the midrib and primary veins of the abaxial leaf surfaces in the form of a non-glandular trichome with smooth surfaces, which is approximately 367.6 µm long. The surface roughness in *Q. acutissima* was approximately 1.7-fold higher on the abaxial leaf surfaces than on the adaxial surfaces (Figure 6b). The stomata of the hypostomatic type had a length of 17.5 µm and a stomatal density of 600.4 per mm$^2$. Non-glandular trichomes were present on both leaf veins and had a relatively higher density over the entire abaxial surface. Specifically, the adaxial surfaces of *Q. acutissima* showed the presence of sessile fasciculate trichomes on the veins. The abaxial and adaxial leaf surfaces of *M. glyptostroboides* showed the development of thick layers of epicuticular wax

(Figure 6c). The leaves were hypostomatic and exhibited a stomatal length of 22.9 μm and stomatal density of 233.2 per mm². The abaxial leaf surfaces of *E. japonicus* had a greater surface roughness than the adaxial leaf surfaces (Figure 6d). The stomata of the hypostomatic type had a length of 20.2 μm and a stomatal density of 247.7 per mm². *P. densiflora* showed similar surface roughness on adaxial and abaxial needles (Figure 6e). The leaves were amphistomatic (i.e., stomata are present on both leaf surfaces) and the stomatal lengths on the abaxial and adaxial sides were approximately similar at 31.2 μm and 33.9 μm, respectively. Stomatal density was higher on abaxial surfaces compared to adaxial needle surfaces, with values of 110.6 per mm² and 83.1 per mm², respectively.

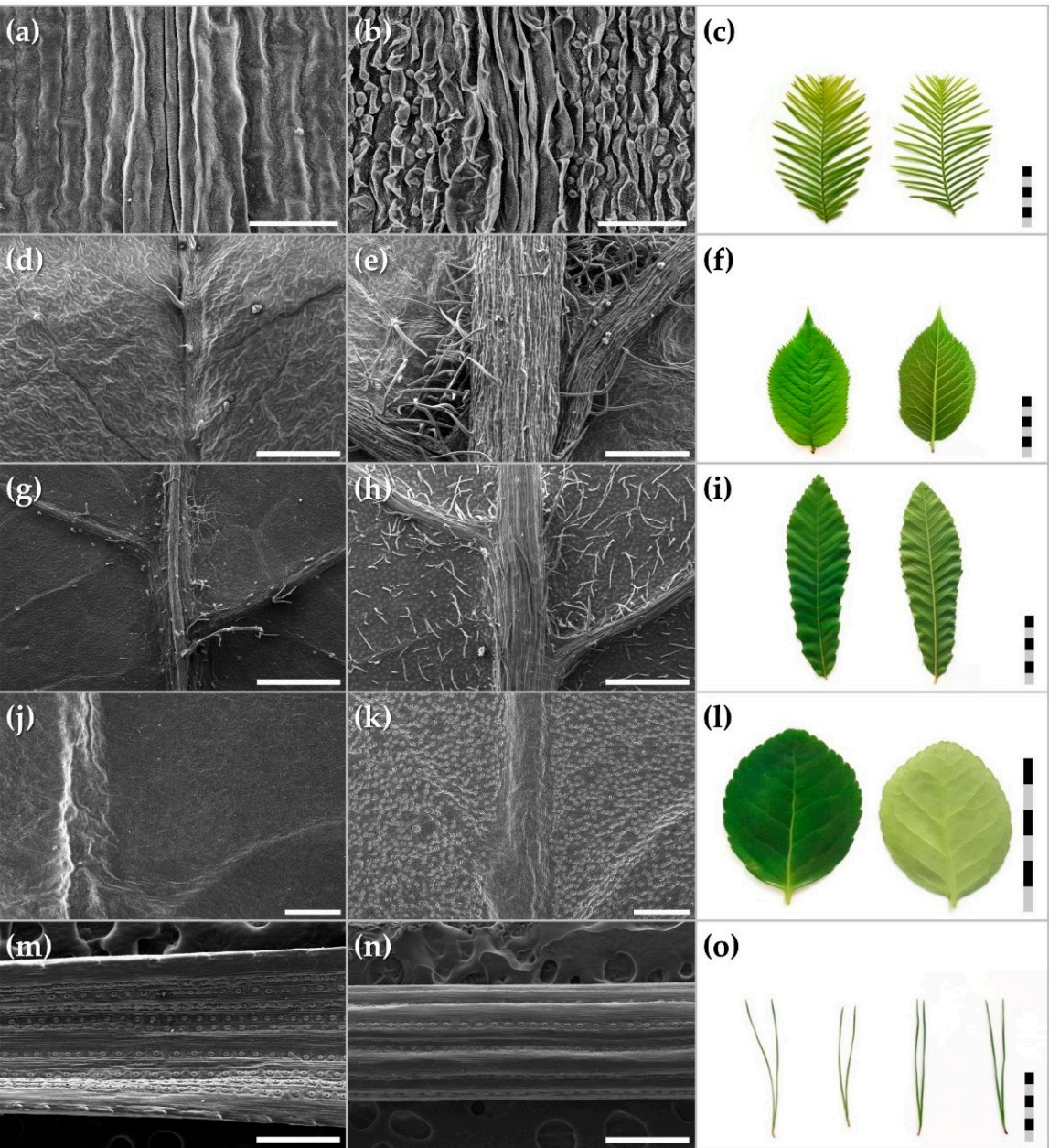

**Figure 4.** Representative micromorphological and leaf morphological variations on adaxial (**a,d,g,j,m**) and abaxial (**b,e,h,k,n**) leaf surfaces of (**a–c**) *M. glyptostroboides*, (**d–f**) *P. yedoensis*, (**g–i**) *Q. acutissima*, (**j–l**) *E. japonicus*, and (**m–o**) *P. densiflora*. Scale bars: (**d,e,g,h,j,k,m,n**) 500 μm; (**a,b**) 100 μm; (**c,f,i,l,o**) 5 cm.

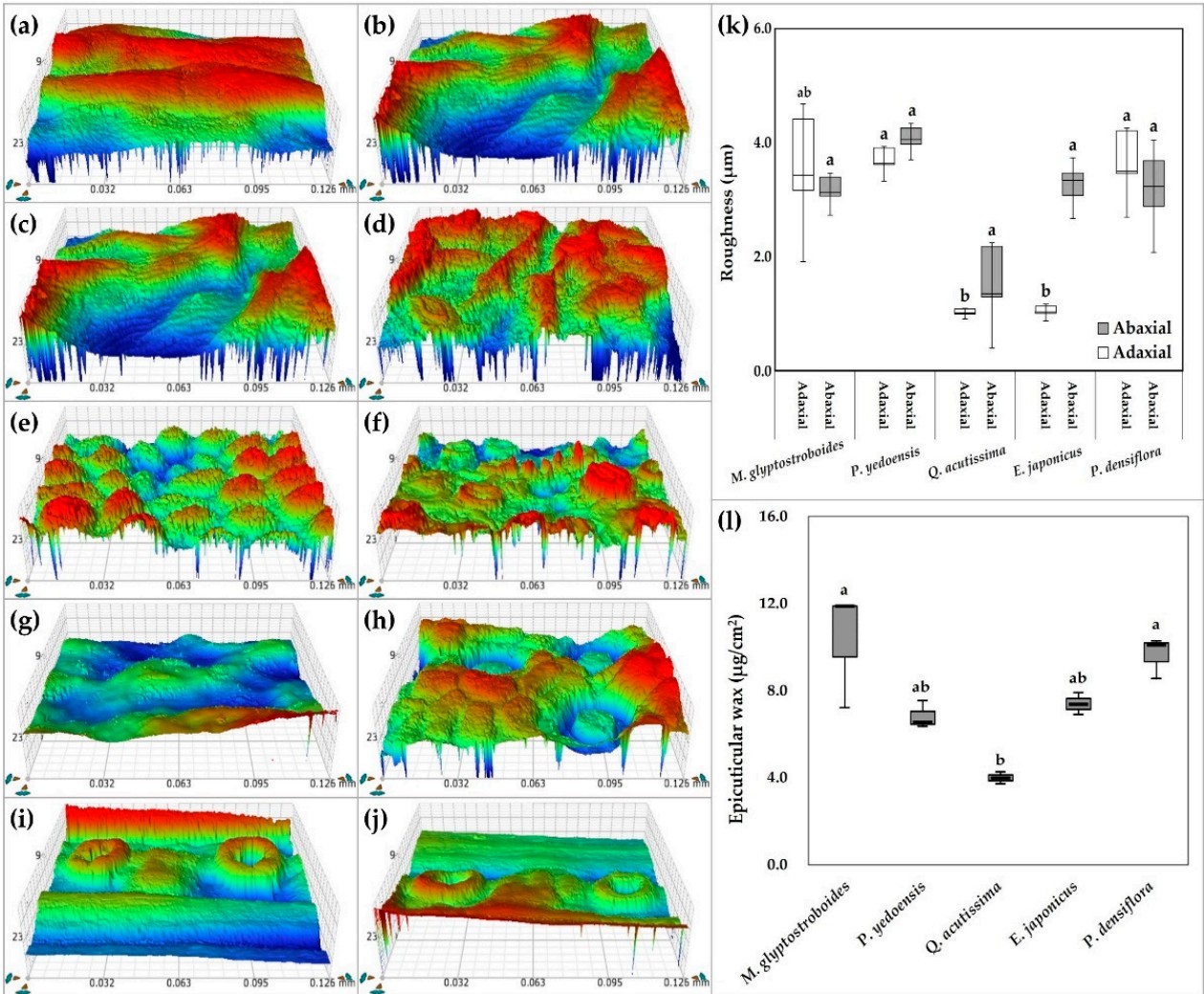

**Figure 5.** 3D surface topography variations of leaf surfaces among different species, which are distinguished by roughness and epicuticular wax: (**a–j**) 3D surface topography reconstructions showing surface roughness using non-contact surface profiler on (**a,c,e,g,i**) adaxial and (**b,d,f,h,j**) abaxial leaf surfaces (magnification: ×50), (**k**) leaf roughness across adaxial and abaxial leaf surfaces, and (**l**) cuticular wax quantification. Box plots depict the median value (centerline) and the first and third quartiles (box edges). Lowercase letters on top of box plots indicate significantly different groups as determined via the non-parametric Kruskal–Wallis test (*p* < 0.05) among species. Note: (**a,b**) *M. glyptostroboides*, (**c,d**) *P. yedoensis*, (**e,f**) *Q. acutissima*, (**g,h**) *E. japonicus*, and (**i,j**) *Pinus densiflora*.

In this study, we profiled the major chemical categories and toxic heavy metals of PM particles on leaf surfaces of five tree species at Seoul Forest Park (Figure 6f,g). As shown in Figure 6, our results confirmed that fine and coarse particles were present on the adaxial leaf roughness surfaces, which may be mainly caused by the unevenness of the leaf surface. Several PM particles were particulates with toxic metals such as cadmium (Cd), lead (Pb), nickel (Ni), mercury (Hg), and copper (Cu). Detection of these elements showed the presence of different types of irregular aggregates, soot particles, and spherule types, carrying significant contributions from both anthropogenic and natural sources. Coarse particles were classified as aluminosilicate/silica minerals with irregular forms originating from windblown dust from the atmosphere via resuspension from sand, road dust, and construction. Based on the energy spectrum, fine particles were identified as spherical Si-, Al-, and Fe-rich particles, originating from track out or windblown dust from a nearby ready-mix concrete plant. In the present study, the SEM-EDS elemental analysis revealed

that the PM particles adsorbed on leaf surfaces of major trees in urban forests can be divided into different types of irregular aggregates, soot particles, and spherule types from natural source regions and ready-mix concrete plants. PM particle morphological aspects were represented via physical–chemical complex particulate deposition with spherules with relatively smooth surfaces and aggregate particles with coarse surface textures.

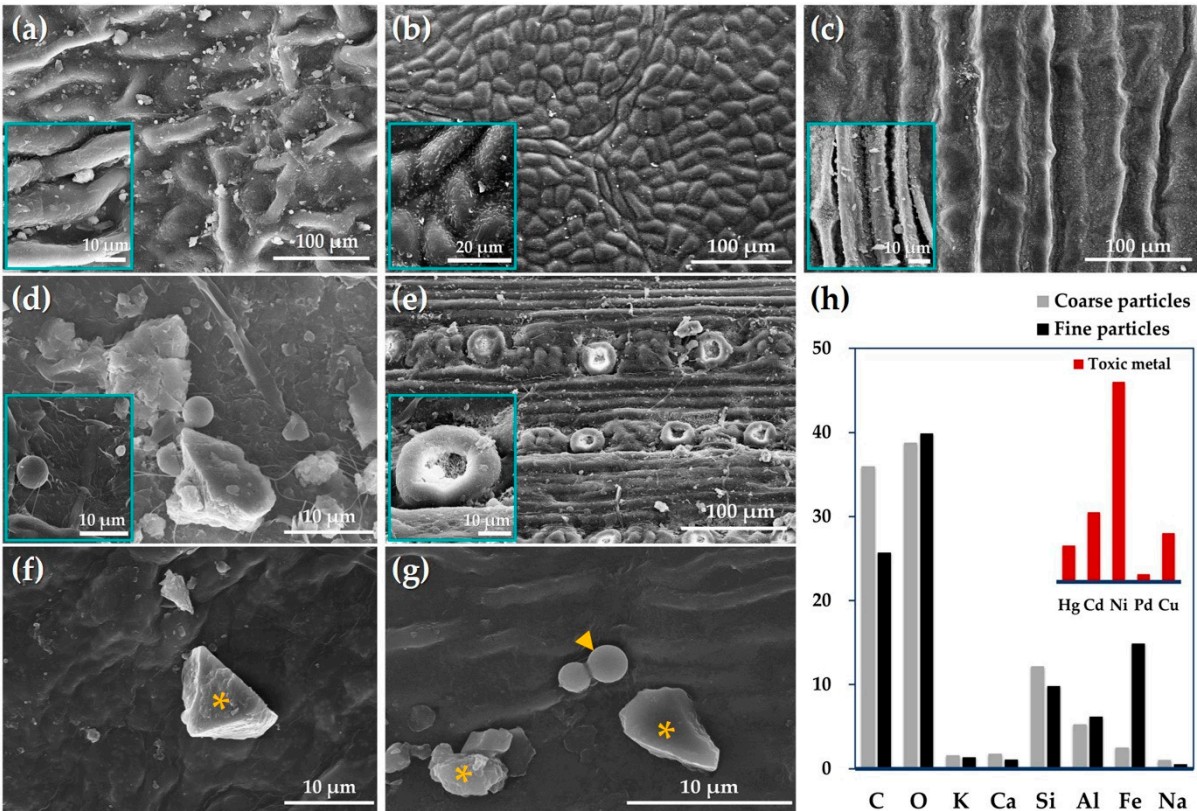

**Figure 6.** Comparative analysis of scanning electron microscopy with energy dispersive X-ray spectroscopy (SEM-EDS) on adaxial leaf surfaces. (**a**) *P. yedoensis* (roughened stripe-like grooves and non-glandular trichomes on both adaxial and abaxial surfaces), (**b**) *Q. acutissima* (adaxial surfaces with bullate-like grooves and abaxial surfaces with simple uniseriate and stellate trichomes), (**c**) *M. glyptostroboides* (waxes on adaxial surfaces), (**d**) *E. japonicus* (smooth adaxial surfaces with thick wax layers and adaxial leaf surfaces with concave stomata), (**e**) *P. densiflora* (elliptical-rounded stomata on both adaxial and abaxial needle surfaces), (**f**,**g**) size-segregated PM particles (e.g., asterisk and arrow) on leaf surfaces used for elemental composition test, and (**h**) SEM-EDS elemental analysis. Note: Images show different surface roughness configurations of grooves, stomata, and particles.

### 3.4. Correlations among Leaf Macro-Scale, Micro-Scale, and Geometric Properties and Net PM Wash-Off Ability from Leaves during Rainfall Events

The correlation between the net PM wash-off ability from the inner and outer crown-positioned leaves and leaf macro-scale, micro-scale, and geometric properties is shown in Figures 7 and 8. In the macro-scale of leaves (Figures 7a and S3), outer SPM2.5 revealed statistically significant negative correlations with LA ($r = -0.73$, $p < 0.01$), circularity ($r = -0.66$, $p < 0.01$), W/L ratio ($r = -0.63$, $p < 0.05$), and V/B ratio ($r = -0.59$, $p < 0.05$). In addition, outer SPM10 showed a strong positive correlation with W/L ratio (0.78, $p < 0.01$) and a negative correlation with V/B ratio ($r = -0.58$, $p < 0.05$). Importantly, outer WPM2.5 had statistically strong positive correlations with RI ($r = 0.66$, $p < 0.01$), while outer WPM10 had no significant correlation with all macro scales of leaves. We found that outer TPM revealed statistically significant positive correlations with W/L ratio ($r = 0.74$, $p < 0.01$) and RI ($r = 0.53$, $p < 0.05$) and showed negative correlations with V/B ratio ($r = -0.61$, $p < 0.05$).

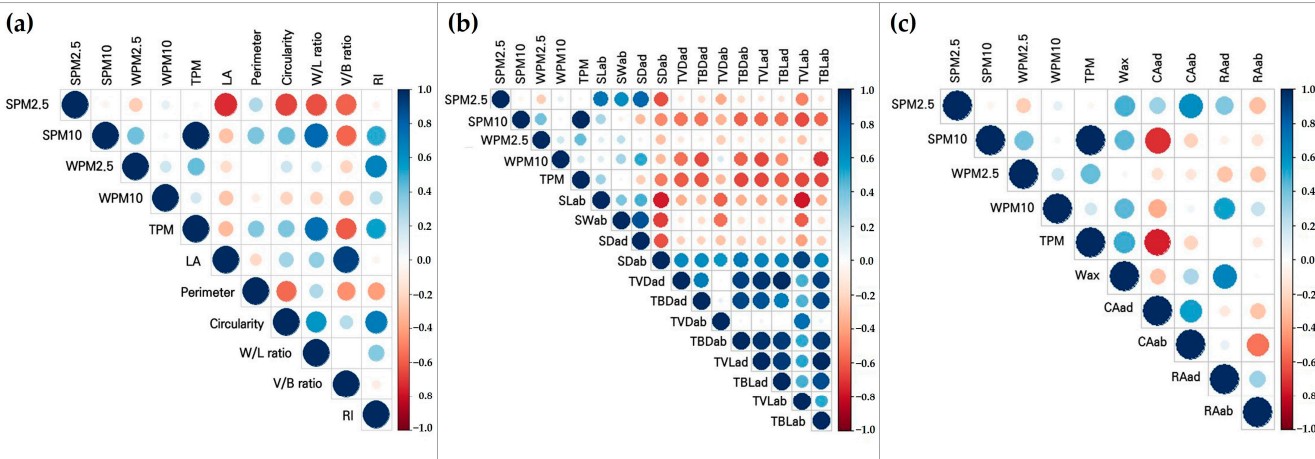

**Figure 7.** Correlation matrix of leaf structural parameters for (**a**) macro-scale, (**b**) micro-scale, and (**c**) geometric properties in the net PM wash-off ability obtained from the outer crown-positioned leaves during rainfall events in the Seoul Forest Park. Data are mean, $n = 10$. Note: The color of the spectrum bar represents the correlation between two variables; red signifies a negative correlation and blue denotes a positive correlation. The correlations are proportional to the color saturation and circle size. A small circle displays a weak correlation, while the blue diagonal denotes autocorrelation.

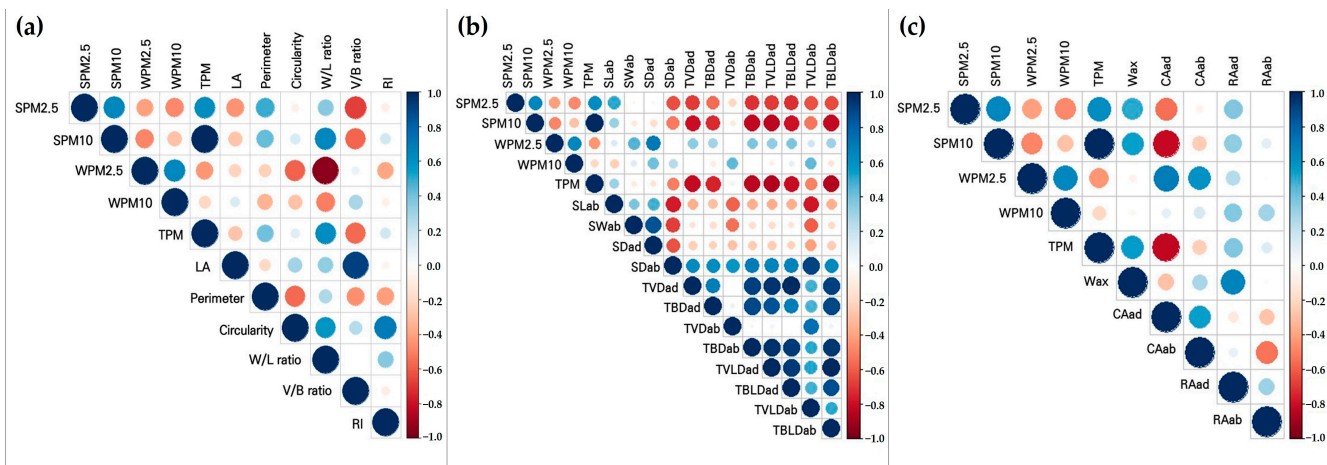

**Figure 8.** Correlation matrix of leaf structural parameters for (**a**) macro-scale, (**b**) micro-scale, and (**c**) geometric properties in the net PM wash-off ability obtained from the inner crown-positioned leaves during rainfall events in the Seoul Forest Park. Note: The color of the spectrum bar represents the correlation between two variables; red signifies a negative correlation and blue denotes a positive correlation. The correlations are proportional to the color saturation and circle size. A small circle displays a weak correlation, while the blue diagonal denotes autocorrelation.

As shown in Figure 8a and Figure S6, the statistical analysis revealed a strong negative correlation between inner SPM2.5 and V/B ratio ($r = -0.68$, $p < 0.01$). Inner SPM10 had a strong positive correlation with the W/L ratio ($r = 0.65$, $p < 0.01$) but had a significant negative correlation with the V/B ratio ($r = -0.58$, $p < 0.05$). Inner WPM2.5 had significantly negative correlations with the W/L ratio ($r = -0.95$, $p < 0.01$) and circularity ($r = -0.59$, $p < 0.05$), while inner WPM10 had a non-significant correlation with all macro scales of leaves. Notably, inner TPM was positively correlated with the W/L ratio ($r = 0.62$, $p < 0.05$) but negatively correlated with the V/B ratio ($r = -0.57$, $p < 0.05$).

According to the correlation matrix analysis above in the case of micro-scale (Figures 7b and 8b), outer SPM10, outer WPM10, and outer TPM showed negative correlations with trichome density and length during rainfall events, while no significant

correlations were present in outer SPM2.5. Interestingly, outer SPM2.5 only revealed significant correlations with stomatal length and width. No significant correlation between outer WPM2.5 and all micro-scale variables on leaf surfaces was found during rainfall events (Figure 7b and Figure S4). Inner SPM2.5, inner SPM10, and inner TPM revealed strong negative correlations with trichome density and length. Importantly, a strong positive correlation was represented between only inner WPM2.5 and stomatal density on adaxial leaf surfaces ($r = 0.71$, $p < 0.01$). Furthermore, inner WPM10 exhibited no significant correlations with all micro-scale variables (Figures 8b and S7).

From a geometric point of view (Figures 7c, 8c, S5 and S8), analyses revealed that Wax was positively correlated with inner SPM10 ($r = 0.55$, $p < 0.05$) and inner TPM ($r = 0.56$, $p < 0.05$) and not for the outer crown-positioned leaves. CAad has significantly negative correlations with outer SPM10 ($r = -0.72$, $p < 0.01$) and outer TPM ($r = -0.76$, $p < 0.01$), and CAab has a positive correlation with outer SPM2.5 ($r = 0.62$, $p < 0.05$). The inner WPM2.5 revealed a positive correlation coefficient with CAad ($r = 0.68$, $p < 0.01$) and CAab ($r = 0.59$, $p < 0.05$). No significant correlation was observed for the RAad or the RAab with all PM removal ability variables, except for the outer WPM10. Our study found positive correlations between RAad and outer WPM10 ($r = 0.53$, $p < 0.05$).

## 4. Discussion

Urban vegetation and building surfaces can improve air quality by blocking and retaining airborne PM pollution [13]. Some PM on the leaves can be encapsulated in the cuticle [28]. Previous studies [11,12] suggested that most PM particles on leaf surfaces are only temporarily retained on leaf surfaces before being resuspended in the air by winds or deposited on the soil by rainfall. Ould-Dada and Baghini [28] suggested that resuspension of some particles can occur in all layers of the tree canopy. Particles deposited in tree canopy locations with low wind speeds are less susceptible to resuspension than particles located at the top of the canopy with high wind speeds. Wind speed can vary by tree height, thus being higher at the top and bottom of the tree canopy and lower in the middle. Unlike wind, rainfall factors can remove PM particles on leaf surfaces through wash-off. Wash-off has a sedimentation effect that transports PM particles to the ground and deposits them on the soil [28]. This represents the net wash-off ability of PM particles [12,13] from the atmosphere by urban plants. Nonetheless, soils can become their potential sink and source for air pollutants, including their precursors. Additionally, soils enable plant development, which is crucial for controlling air quality. The magnitude of the effects of soil on air quality ranges from local to global, driving the local environment and global climate [29].

Very few studies have examined substantial dynamics in PM retention and wash-off on leaves relative to inner and outer tree crowns in different rainfall intensities. In this study, we found that the PM removal from plants across different rainfall intensities was highest for particle size $PM_{10}$. Although most tree species tested under different rainfall intensities showed differences in PM retention and wash-off between inner and outer crown-positioned leaves, heavy rainfall intensity of 19.5 mm/h for 3 h showed the highest particle removal for $PM_{10}$ compared to that for $PM_{2.5}$. This indicates that the wash-off effect by rainfall is higher for $PM_{10}$ than for $PM_{2.5}$, and this difference widens with increasing rainfall intensity [15,16].

Trees that comprise a large portion of urban forests can play a vital role as the first line of defense to improve air quality by intercepting atmospheric noxious pollutants (i.e., harmful gases and particulates) by tree canopies such as *Cassia fistula*, *Ficus benghalensis*, *Ficus elastica*, *Ficus rubra*, *Magnolia grandiflora*, *Pinus pinea*, *Platanus × acerifolia*, and *Tectona grandis* [29,30]. Densely distributed trichomes and roughness of adaxial leaf surfaces may be the main reasons for the improvement in PM retention capacity compared with abaxial surfaces [24]. $PM_{2.5}$ or $PM_{10}$ may be retained or absorbed by stomata [31]. The analysis of PM mass deposited on leaf surfaces showed that PM particles up to 2 μm in diameter are present in the stomata [32,33]. As shown in Figure 6e, the attachment of PM particles was confirmed inside the inner wall

of stomata or the periphery of guard cells of evergreen species, which have stomata that are easily distinguishable from other tree species.

The ability of plants to function as adsorbents for PM particles is generally significantly correlated with leaf surface microstructural characteristics such as leaf roughness, leaf area, roughness, grooves, trichomes, and stomatal density [5,21,34–37]. The influence of weather factors (especially rainfall) on PM adsorption and desorption has been mentioned in many studies [13–16,24,38], and rainfall intensity and duration have an interaction effect with the amount of PM adsorption. Figures 1–3 show that rainfall intensity has different PM retention and wash-off abilities for particle size levels. On the other hand, if there is no PM wash-off from leaf surfaces, the capacity for PM adsorption on the leaf surfaces is saturated, which may lead to lower PM adsorption efficiency [22]. In addition, the ability to adsorb PM on leaf surfaces can be renewed with the wash-off process [12,17]. PM particles undergo a process of being washed off into the soil with rainfall, and these processes have been proposed as an effective method to refresh PM accumulation capacity [16,22].

Previous studies have highlighted the important correlation between leaf-trait combinations and PM adsorption [39,40]. PM particles of different sizes may exhibit different levels of adhesion to leaf surfaces. Large particles are loosely attached to the leaf surface of the plant and are not hard to contact, so they are easily washed away by rainwater [38]. As such, the PM wash-off effect on leaf surfaces of trees by rainfall is affected by particle size characteristics, which depend on leaf surface ultrastructure, leaf area tree species, physical factors, plant community, land use, etc. Furthermore, it has been demonstrated that the retention rate of particulates is negatively correlated with the microroughness of leaf surfaces and the amount and composition of epicuticular waxes [41]. Another important aspect is the hydrophobic properties of higher epicuticular wax in the leaf tissues; the contact area between the particles and the leaf surfaces might be considerably reduced because of their low affinity [42]. Plant species differ in the amount, composition, and structure of epicuticular waxes on leaf surfaces, which has practical implications for particle trapping [39,43].

Plant species have a direct impact on PM deposition and retention capacity of leaf surfaces [39,40,44] owing to the variability of plant functional traits by species-specific phenological shifts [9,45]. Notably, Fusaro et al. [46] demonstrated that plant functional groups such as morphological, physiological, structural, and phenological traits that influence the fundamental drivers for plant performance can play a pivotal role as mediators of phytoremediation and biomonitoring approaches in urban areas. Various industrial processes and vehicular traffic produce harmful emissions of magnetic minerals and heavy metals. Shah et al. [47] showed that morphological, biochemical, and physiological responses in plants are significantly influenced in numerous ways by cement factory-derived pollutants comprising cement dust. Significantly, cement dust deposition in urban forests can affect soil properties, plant development, and human health. Figure 6 shows fine and coarse particles of different origins with size measurements. Anthropogenic pollution that can lead to the Si-Al-rich fly ash includes burning fossil fuels, burning biomass, and smelting [6]. In the present study, the SEM-EDS elemental analysis revealed that the PM particles adsorbed on leaf surfaces could be divided into irregular aggregates, soot particles, and spherule types originating from natural sources and anthropogenic ready-mix concrete plants. The spherical Fe-rich particles can be originated artificially in coal boilers, metal industries, and power plants with different temperature conditions, gradually accumulating on leaf surfaces via the resuspension process of the soil already mentioned above.

Natural rainfall does not necessarily function as a particulate wash-off [16,22]. Larger coarse PM particles deposited on leaf surfaces have greater wash-off levels than fine PM under heavy rainfall intensity [11,13]. Importantly, the present study showed that no definite levels could be found, indicating wash-off effects of in-wax PM for two size fractions by different rainfall intensities in both inner- and outer-crown needles of *P. densiflora*. Broadleaved trees can show a higher $PM_{2.5}$ wash-off effect compared to coniferous trees, and high rainfall intensity can shorten the $PM_{2.5}$ cycle of leaves and show a high wash-off effect [22]. Some studies have shown

that appropriate planting designs are more significant than tree species selection in decreasing ambient PM concentrations in urban contexts [48,49]. Nevertheless, numerous studies have identified that it is important to consider the selection of appropriate tree species in attenuating the ambient PM concentrations in urban areas. Tree species with leaf surface microstructures, such as epicuticular wax, hair/trichomes, and surface ridges are better for reducing airborne PM. However, when comparing PM net removal rates, urban plants with smooth leaves are generally less effective accumulators of PM, but positively affect the efficiency of PM wash-off events [5–7,9,13,35].

During the rainfall event, remarkably, we found apparent PM retention and/or new PM capture due to a drastic increase of PM particles (especially $PM_{2.5}$) at inner crown-positioned leaves of *Q. acutissima* accompanied by micro-level surface roughness with dense and narrow grooves. The rainfall event was generally found to increase the wash-off levels of SPM10 (Figure 1), except for *Q. acutissima* (Figure 1b), which showed significant SPM2.5 and SPM10 retention levels in inner-crown leaf surfaces. However, in this study, *Q. acutissima* had no significant wash-off of PM particles encapsulated within wax according to tree crown positions during rainfall events (Figure 2b) because of their relatively low wax contents (Figure 5l).

In the seasonal dry or wet depositions, microparticles remain adsorbed to the leaves, and resuspension to the atmosphere may be less likely under normal weather conditions [50,51]. After certain periods, the PM adsorption by the plant leaves reaches saturation and leaves no longer adsorb fine dust particles. PM particles that remain saturated on the leaf surface for a long time harm leaves and affect photosynthetic efficiency [21,38]. In high rainfall intensity, PM wash-off was significantly increased in all species except *Q. acutissima* (Figure 1). Notably, *E. japonicus* and *M. glyptostroboides* also showed high levels of PM wash-off effect (especially in SPM and TPM) in the entire tree crown based on the LAI. Based on the KMA, rainfall intensity is generally classified into four categories of light (<3 mm/h), moderate (3–15 mm/h), heavy (15–30 mm/h), and violent (>30 mm/h). Previous studies generally consider rainfall duration, pattern, and intensity as a function of rainfall characteristics to PM retention and wash-off dynamics [7,15]. These results suggest that PM retention and wash-off ability depended on rainfall intensity and plant species. Moreover, Xu et al. [52] demonstrated that it is limited in real rainfall by its lower intensity and distribution within the tree canopy. Thus, based on these findings, we only considered heavy rainfall to understand the effects of rainfall in relation to leaf traits. Therefore, the PM wash-off could be a likely evident during the summer rainy season in high-intensity rainfall, which frequently occurs from mid-June to the end of July. In contrast, its effects could be even low during low–high-intensity rainfall, which occurs from autumn to spring. Leaves of *E. japonicus*, the evergreen broadleaf tree species, could play an essential role in PM uptake in city streets, primarily because of their soft, thick, and leathery adaxial surfaces and abaxial leaf surfaces with sunken stomata, increasing their ability to capture PM [25,53]. More importantly, *E. japonicus* has a high ability to adsorb PM particles in the air throughout periods of limited rainfall and is effective in the PM wash-off from leaves with raindrops during rainfall events, cleaning leaf surfaces and depositing the PM into the soil (Figure 1). In such situations, it is possible to revest their leaf surface areas that can newly deposit PM pollution during rainfall seasons.

Differences in epicuticular wax ultrastructures, such as thin films, platelets, and tubules [54], show considerable potential in capturing PM during rainfall. Furthermore, regarding the regeneration of wax from leaves, although there may be a significant difference in the rate of wax regeneration, the regeneration of damaged wax structures seems clear. In particular, the wax regeneration was confirmed after six days [55]. From these results, PM particles encapsulated in the epicuticular wax can be removed entirely into the soil without resuspending into the atmosphere due to rainfall events. Leaf surfaces allow for dry and wet depositions depending on their affinity for PM that may be able to immobilize within epicuticular wax layers, trichomes, or inner substomatal cavities [56]. PM retention and wash-off by falling raindrops are classified as wet deposition, and atmospheric PM is markedly reduced after rainfall events. Wang et al. [16] demonstrated that PM particles car-

ried by raindrops become dust patches on the leaf surface after the moisture evaporates and can be visually checked. In general, rainfall conditions can effectively desorb PM particles from leaf surfaces, and it is known that relatively higher rainfall intensity is more effective in removing air pollutants. New particles suspended in the atmosphere accumulate on the leaf surfaces after rainfall. This study showed that higher rainfall intensity promoted the wash-off effect of PM particles from leaf surfaces. In addition, it was shown that low rainfall intensity promotes the accumulation of PM particles on leaf surfaces due to wet deposition. This contradictory phenomenon was also found in a study of seasonal changes in the deposition of PM particles on leaf surfaces [57]. Despite differences in micro and macro morphological traits, five species in this study showed potential to ensure surface areas that can newly trap atmospheric particles due to the PM wash-off from leaf surfaces under high rainfall intensity [11,12].

The multi-faceted nature of trees' impacts on the surrounding environment is complicated and confounded by species-specific traits, phenomena, and exogenous stressors [58]. Raindrops blocked by leaves or branches in tree crowns during rainfall are temporarily stored on the surface and evaporate with the rain stopping [59]. Likewise, trees are crucial for improving the water cycle (i.e., urban hydrology) in urban areas by blocking raindrops to increase evaporation and reduce runoff. Depending on coniferous and broad-leaved species, throughfall drops classified as free throughfall, splash throughfall, and canopy drip are different. [58–60]. Throughfall may be classified into two categories, free and release, with release throughfall further divided into splash throughfall and canopy drip [60]. Notably, plant surface, weather, canopy conditions, and biophysical properties have a role in determining the diameter of the canopy drip [58]. Yang et al. [59] evaluated the effect of crown shape on rain interception using main street trees in Seoul, and the average interception rate during rainfall events according to species was different according to tree species as follows: *Ginkgo biloba* (57.93%), *Sophora japonica* (35.79%), *Aesculus turbinata* (30.58%), and *Zelkova serrata* (20.59%). Yang et al. [59] also revealed that small leaves were more effective at intercepting rainfall than large leaves and showed that rainfall intensity was related to rainfall interception. Moreover, the differences in rainfall interception rates between the tree species were more significant for low-intensity rainfall since small rainfall events were insufficient to saturate the tree canopies with a large storage capacity [59].

## 5. Conclusions

Our field study uncovers the vital role of leaf macro-scale, micro-scale, and geometric features concerning structural descriptors for PM wash-off dynamics during rainfall events in leaf samples taken from the inner and outer tree crowns of tree species. During rainfall events, there were significant differences among tree species in particle size fractions for inner and outer crown-positioned leaves. In summary, rainfall intensity affects the PM wash-off in inner and outer tree crowns. In particular, the rainfall intensity is lower in the inner crowns than in the outer crowns. Inner crown-positioned leaves suppressed PM wash-off via cuticle hydration due to decreased rainfall intensity. In general, the wash-off levels of different-sized PM particles from leaves (especially $PM_{10}$ deposited on leaf surfaces) on the ground increased after rainfall, except in *Q. acutissima*. Different tree species showed variable response potential for PM wash-off and/or retention during rainfall; *E. japonicus* and *M. glyptostroboides* without trichomes were considerably higher in PM wash-off levels on leaf surfaces. In contrast, *Q. acutissima* with micro-level surface roughness with dense and narrow grooves showed the lowest PM wash-off levels. More importantly, *E. japonicus* could play an essential role in removing PM pollution adsorbed on leaves during the descent of raindrops to the ground, cleaning leaf surfaces, and depositing the PM into the soil. Correlations between several leaf surface traits and PM particle size fractions showed a vital transition between inner and outer crown-positioned leaves. Notably, in high rainfall intensity, PM wash-off increases with leaf turbulence, and this feature is highly dependent on macro- (W/L ratio and roughness index) and micro-structure (stomatal density) and geometric properties (wax and contact angle) of leaf surfaces. Our findings indicate slight

correlations with the microstructure in the PM wash-off of large particles during rainfall, whereas strong correlations were presented with small particles. Therefore, PM wash-off effects by individual tree crowns can be helpful for a comprehensive understanding of urban trees, atmospheric quality reduction, and urban hydrology in improved management and planning of urban forests and urban green areas.

**Supplementary Materials:** The following supporting information can be downloaded at: https://www.mdpi.com/article/10.3390/horticulturae9020165/s1, Figure S1: Percentage changes of PM retention (+) and wash-off (−) in PM mass on both leaf surfaces and in waxes under three rainfall intensities; Figure S2: Representative scanning electron microscopy images showing various leaf microstructures and surface roughness on (a,b,e,f,i,j,m,n,q,r) adaxial and (c,d,g,h,k,l,o,p,s,t) abaxial leaf surfaces; Figure S3: Correlation matrix for all pairs of the observed variables between leaf macro-scale and net PM wash-off ability from the outer crown-positioned leaves; Figure S4: Correlation matrix for all pairs of the observed variables between leaf micro-scale and net PM wash-off ability from the outer crown-positioned leaves; Figure S5: Correlation matrix for all pairs of the observed variables between leaf geometric properties and net PM wash-off ability from the outer crown-positioned leaves; Figure S6: Correlation matrix for all pairs of the observed variables between leaf macro-scale and net PM wash-off ability from the inner crown-positioned leaves; Figure S7: Correlation matrix for all pairs of the observed variables between leaf micro-scale and net PM wash-off ability from the inner crown-positioned leaves; Figure S8: Correlation matrix for all pairs of the observed variables between leaf geometric properties and net PM wash-off ability from the inner crown-positioned leaves; Table S1: Quantitative assessment of retained PM and wash-off on leaf surfaces and wax layers during rainfall events; Table S2: Leaf macro-scale properties of individual leaf shape descriptors in urban forest tree species; Table S3: Leaf micro-scale structural properties of individual leaf surfaces in urban forest tree species; Table S4: Leaf geometric properties in urban forest tree species.

**Author Contributions:** Conceptualization, S.Y.W.; methodology, M.J.K. and J.L; validation, J.-a.S., S.M.J., H.C. and C.-Y.O.; formal analysis, M.J.K., S.P., and J.L.; investigation, S.P., H.K., Y.J.L. and S.G.J.; resources, J.L.; data curation, Y.J.L., S.P. and H.K.; writing—original draft preparation, M.J.K. and J.L.; writing—review and editing, M.J.K.; visualization, M.J.K.; supervision, S.Y.W.; project administration, J.L.; funding acquisition, S.Y.W. and K.K. All authors have read and agreed to the published version of the manuscript.

**Funding:** This study was carried out with the support of 'A Study on Mechanism and Function Improvement of Plants for Reducing Air Pollutants' (Grant No. FE0000-2018-01-2021) from the National Institute of Forest Science (NIFoS), Republic of Korea.

**Data Availability Statement:** The data presented in this study are available on request from the corresponding author. The data are not publicly available due to privacy or other restrictions.

**Acknowledgments:** Special thanks go to the Center for Research Facilities at the University of Seoul for the microscope technical support in the experimental field.

**Conflicts of Interest:** The authors declare no conflict of interest.

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
