# Peer review of "Understanding Particulate Matter Retention and Wash-Off during Rainfall in Relation to Leaf Traits of Urban Forest Tree Species"

_horticulturae, doi:10.3390/horticulturae9020165_

Round 1
Reviewer 1 Report
1. In the Introduction section: it is better if you add more information related to the tree species that already used to reduce particulate matter in the urban areas (from the previous studies).
2. In the Methodology section: please also describe how did you prevent the wind (wind speed) during sample collection because the wind speed also influence the particulate matter retention in the leaf surface.
3. In the Discussion section: please add discussion related the application of your results; based on the leaf macro-scale, micro-scale, and geometric features, what kind of tree species that you recommend being planted in the urban area to reduce particulate matter.
Author Response
Author Reply to the Review Report
(Manuscript ID: horticulturae-2110117)
Dear Editor,
Title
Understanding particulate matter retention and wash-off during rainfall in relation to leaf traits of urban forest tree species
We are pleased to resubmit for publication the revised version of Manuscript ID: horticulturae-2110117 entitled “Understanding particulate matter retention and wash-off during rainfall in relation to leaf traits of urban forest tree species”. We are very grateful to the Associate Editor and three Reviewers for their deep and detailed comments which have helped us to improve our manuscript. We have revised our manuscript according to the suggestions and comments of three anonymous reviewers. Grammatical error correction and English improvement were carried done by a native-speaking English editor as suggested. Please find below point-by-point responses to each of the reviewers’ comments on our manuscript. Upload files of revised manuscripts were attached in two forms: ‘Track Changes’ and ‘Final PDF version’. Finally, we hope that reviewers and editors will be satisfied with the response to reviewer comments.
Sincerely yours,
Response to Reviewer 1 Comments
Point 1: In the Introduction section: it is better if you add more information related to the tree species that already used to reduce particulate matter in the urban areas (from the previous studies).
Response 1: As per your requirement, we introduced some trees with the PM capturing performance in urban tree species in the 'Introduction' section.
In the ‘Introduction’ section (lines 43-47)
Urban trees can be widely used as biological filters that intercept airborne PM (e.g., Buxus koreana [6], Cedrus deodara [5], Euonymus japonicus [6,8], Pinus tabulaeformis [5,7], Sophora japonica [7,8], Taxus cuspidata [6], and Ulmus pumila [5,8]). Previous studies have demonstrated that PM capturing and retaining capacities on leaves depend on surface roughness and microstructure properties [5–10].
Point 2: In the Methodology section: please also describe how did you prevent the wind (wind speed) during sample collection because the wind speed also influence the particulate matter retention in the leaf surface.
Response 2: Therefore, we conducted sampling at the middle part of the tree canopy, which is less susceptible to PM resuspension by wind speed than at the top and bottom of the tree canopy with high wind speeds, to analyze PM residual and wash-off on leaves before and after rainfall events.
In the ‘Discussion’ section (lines 480-484)
As Ould-Dada and Baghini [28] suggested, the resuspension of some particles can occur in all layers of the tree canopy. Particles deposited in tree canopy locations with low wind speeds are low are less susceptible to resuspension than particles located at the top of the canopy with high wind speeds. Wind speed can vary by tree height, thus being higher at the top and bottom of the tree canopy and lower in the middle.
Point 3: In the Discussion section: please add discussion related the application of your results; based on the leaf macro-scale, micro-scale, and geometric features, what kind of tree species that you recommend being planted in the urban area to reduce particulate matter.
Response 3: We already mentioned and recommended plants showing high PM removal capacity in the 'Discussion' section; however, according to the reviewer's suggestion, we added the sentence to emphasize again as follows. Please see lines 590 in the 'Discussion' section.
In the ‘Discussion’ section (lines 590-605)
In high rainfall intensity, PM wash-off were significantly increased in all species except Q. acutissima (Figure 1). Notably, E. japonicus and M. glyptostroboides also showed high levels of PM wash-off effect (especially in SPM and TPM) in the entire tree crown based on the LAI. Based on the KMA, rainfall intensity is generally classified into four categories of light (<3 mm/h), moderate (3–15 mm/h), heavy (15–30 mm/h), and violent (>30 mm/h). Therefore, the PM wash-off could be likely evident during the summer rainy season in high-intensity rainfall, which frequently occurs from mid-June to the end of July. In contrast, its effects could be even low during low-high-intensity rainfall, which occurs from autumn to spring. Leaves of E. japonicus, the evergreen broadleaf tree species, could play an essential role in PM uptake in city streets, primarily because of their soft, thick, and leathery adaxial surfaces and abaxial leaf surfaces with sunken stomata, increasing their ability to capture PM [25,52]. More importantly, E. japonicus has a high ability to adsorb PM particles in the air throughout periods of limited rainfall and is effective in the PM wash-off from leaves with raindrops during rainfall events, cleaning leaf surfaces and depositing the PM into the soil (Figure 1). In such situations, there are possible to revest their leaf surface areas that can newly deposit PM pollution during rainfall seasons.

Reviewer 2 Report
The study by Myeong Ja Kwak et al. is very interesting and covers a very important topic, meaning the particulate matter retention and wash-off during rainfall in relation to leaf traits of urban forest tree species. The manuscript is well-written and easy to read, however some clarification are necessary to consider the paper suitable for publication in this journal. Follows a line by line review:
L41: can you introduce which trees?
L40: you can avoid the word LAND, is superfluous here
L51: what do you mean by surface traits?
L63: in the introduction a huge part about leaf anatomical traits, like stomata, cuticle, hair and their role in dust/PM accumulation and retention is missing.
Please refer to:
10.1111/plb.12966
10.1016/j.jhazmat.2018.02.044
10.1007/s11356-021-13242-9
L66: why? what is the difficulty of conducting experiments in the field
L71: more than microstructure I would say "anatomy" which is more recognized in this kind of studies
L85: per individual?
L102-104: this sentence is not clear to me, can you explain it better?
L108-109: so you only considered heavy rainfall if I understood correctly
L119-121: was this rainfall simulated or real? if real how did you measure that? and how you can discern between rainfall of 40mm and other heavier or lighter rainfall? this is not very clear to me
L132-134 AND THE WHOLE TEXT: as mentioned before is more common to refer to these traits as morphological (macro) and anatomical (micro) traits
L158-159: so the anatomical parameters weren't calculated from the experimental samples?? why is that?
L213-216: this is not true, because for inner there is difference, I'm aware you added the reference to Figure1b which refer to outer but from the sentence it has another meaning, please specify
Figure 1: it is not very clear without reading the caption that graph a is for inner and b for outer leaves. You can write it on the graph.
Figure 2 : I would suggest to add letters for significant difference as the graph before
L462-465: Do you mean that PM which was wash-off and has sedimentation in the soil can be resuspended again and sedimented again on leaves? can you say more about it?
L475: was it true for all the species tested?
L495-499: please avoid repetition
L500: please format the reference in the right way
L594-597: please avoid repeating "in this study". You can add "here"
Author Response
Author Reply to the Review Report
(Manuscript ID: horticulturae-2110117)
Dear Editor,
Title
Understanding particulate matter retention and wash-off during rainfall in relation to leaf traits of urban forest tree species
We are pleased to resubmit for publication the revised version of Manuscript ID: horticulturae-2110117 entitled “Understanding particulate matter retention and wash-off during rainfall in relation to leaf traits of urban forest tree species”. We are very grateful to the Associate Editor and three Reviewers for their deep and detailed comments which have helped us to improve our manuscript. We have revised our manuscript according to the suggestions and comments of three anonymous reviewers. Grammatical error correction and English improvement were carried done by a native-speaking English editor as suggested. Please find below point-by-point responses to each of the reviewers’ comments on our manuscript. Upload files of revised manuscripts were attached in two forms: ‘Track Changes’ and ‘Final PDF version’. Finally, we hope that reviewers and editors will be satisfied with the response to reviewer comments.
Sincerely yours,
Response to Reviewer 2 Comments
The study by Myeong Ja Kwak et al. is very interesting and covers a very important topic, meaning the particulate matter retention and wash-off during rainfall in relation to leaf traits of urban forest tree species. The manuscript is well-written and easy to read, however some clarification are necessary to consider the paper suitable for publication in this journal. Follows a line-by-line review:
First, we greatly appreciate Reviewer 2's comment. Thank you for pointing out some problems. After correcting grammatical errors and improving our English, we revised the manuscript by adding and deleting some sentences and paragraphs.
Point 1: L41: can you introduce which trees?
Response 1: Based on Reviewer 2's comment, we introduced some trees with the PM capturing performance in urban tree species from the previous studies in the 'Introduction' section.
In the ‘Introduction’ section (lines 43-47)
Urban trees can be widely used as biological filters that intercept airborne PM (e.g., Buxus koreana [6], Cedrus deodara [5], Euonymus japonicus [6,8], Pinus tabulaeformis [5,7], Sophora japonica [7,8], Taxus cuspidata [6], and Ulmus pumila [5,8]). Previous studies have demonstrated that PM capturing and retaining capacities on leaves depend on surface roughness and microstructure properties [5–10].
Point 2: L40: you can avoid the word LAND, is superfluous here
Response 2: As per your comment, it was deleted.
Point 3: L51: what do you mean by surface traits?
Response 3: As requested, in the Introduction section, we added brief information about leaf surface traits affecting PM net removal by rainfall.
In the ‘Introduction’ section (lines 58-61)
Other factors that may affect PM removal by rainfall are leaf shapes and surface traits (e.g., smooth surfaces [13], trichome [13], epicuticular wax [13,15], contact angle [15], and groove [15]) when raindrops hit leaf surfaces, the hydrological characteristics of rainfall events, and PM retention mass before the rainfall events [9,12,13,15].
Point 4: L63: in the introduction a huge part about leaf anatomical traits, like stomata, cuticle, hair and their role in dust/PM accumulation and retention is missing. Please refer to: 10.1111/plb.12966; 10.1016/j.jhazmat.2018.02.044; 10.1007/s11356-021-13242-9
Response 4: According to the reviewer's suggestion, we cited the reference [De Micco et al., 2020], which was comprehensively reviewed in the recommended paper, and added it to the introduction.
In the ‘Introduction’ section (lines 54-57)
Plant species growing in polluted areas may undergo morphological changes such as stomata, trichomes, surfaces, and cuticles, affecting overall photosynthesis, stomatal conductance, and transpiration rate. Thus, plants may considerably reduce their essential function as biological filters [14].
Point 5: L66: why? what is the difficulty of conducting experiments in the field.
Response 5: As per your comments, we added and modified the sentence as follows.
In the ‘Introduction’ section (line 71-82)
In many contexts, field experiments are subject to many limitations due to other environmental interference factors such as atmospheric humidity, temperature, and wind speed; thus, the rainfall-induced wash-off processes have been proactively carried out using rainfall simulation experiments [7,17,21,22]. Therefore, there are few available field studies to date. On the other hand, extending the in-chamber seedling experiment to large trees growing in cities cannot be easy; hence, additional field research is required to explore the leaf potential in dynamic PM retention and wash-off processes in tree crowns [8,20].
Point 6: L71: more than microstructure I would say "anatomy" which is more recognized in this kind of studies
Response 6: Generally, many studies use the term "microstructure" when explaining the relationship between fine dust adsorption/removal ability and leaf surface characteristics. We want to consider using it as a 'microstructure' to give unity to the manuscript. Examples can be found in the following references.
References
Chávez-García, E.; González-Méndez, B. Particulate matter and foliar retention: current knowledge and implications for urban greening. Air Qual. Atmos. Health. 2021, 1–22, doi: 10.1007/s11869-021-01032-8.
Guo, L., Ma, S., Zhao, D., Zhao, B., Xu, B., Wu, J., ... & Chang, Z. (2019). Experimental investigation of vegetative environment buffers in reducing particulate matters emitted from ventilated poultry house. Journal of the air & waste management association, 69(8), 934-943.
Kwak, M.J.; Lee, J.K.; Park, S.; Kim, H.; Lim, Y.J.; Lee, K.A.; Son, J.; Oh, C.Y.; Kim, I.; Woo, S.Y. Surface-based analysis of leaf microstructures for adsorbing and retaining capability of airborne particulate matter in ten woody species. Forests 2020, 11, 946 doi: 10.3390/f11090946.
Li, X., Zhang, T., Sun, F., Song, X., Zhang, Y., Huang, F., ... & Shao, F. (2021). The relationship between particulate matter retention capacity and leaf surface micromorphology of ten tree species in Hangzhou, China. Science of The Total Environment, 771, 144812.
Niu, X., Wang, B., & Wei, W. (2020). Response of the particulate matter capture ability to leaf age and pollution intensity. Environmental Science and Pollution Research, 27(27), 34258-34269.
Xu, L., Yan, Q., Liu, L., He, P., Zhen, Z., Duan, Y., & Jing, Y. (2022). Variations of particulate matter retention by foliage after wind and rain disturbance. Air Quality, Atmosphere & Health, 15(3), 437-447.
Yan, G.; Liu, J.; Zhu, L.; Zhai, J.; Cong, L.; Ma, W.; Wang, Y.; Wu, Y.; Zhang, Z. Effectiveness of wetland plants as biofilters for inhalable particles in an urban park. J. Clean. Prod. 2018, 194, 435–443, doi: 10.1016/j.jclepro.2018.05.168.
Zhang, Z., Gong, J., Li, Y., Zhang, W., Zhang, T., Meng, H., & Liu, X. (2022). Analysis of the influencing factors of atmospheric particulate matter accumulation on coniferous species: measurement methods, pollution level, and leaf traits. Environmental Science and Pollution Research, 1-13.
Zhang, L.; Zhang, Z.; Chen, L.; McNulty, S. An investigation on the leaf accumulation-removal efficiency of atmospheric particulate matter for five urban plant species under different rainfall regimes. Atmos. Environ. 2019, 208, 123–132, doi: 10.1016/j.atmosenv.2019.04.010.
Zhang, W.; Zhang, Y.; Gong, J.; Yang, B.; Zhang, Z.; Wang, B.; Zhu, C.; Shi, J.; Yue, K. Comparison of the suitability of plant species for greenbelt construction based on particulate matter capture capacity, air pollution tolerance index, and antioxidant system. Environ. Pollut. 2020, 263, 114615, doi: 10.1016/j.envpol.2020.114615.
Point 7: L85: per individual?
Response 7: Before transferring sampled twigs into the laboratory, we selected five tree individuals of each species.
In the ‘Materials and Methods’ section (lines 102-105)
We collected n = 120 twigs from branches in the four-quadrant directions of the inner and outer tree crowns at a tree height of 3 to 5 m from a total of n = 5 individual trees of each species at the Seoul Forest Park.
Point 8: L102-104: this sentence is not clear to me, can you explain it better?
Response 8: We already described it at the end of section '2.1'. Additionally, you can also gain detailed information about rainfall intensity in South Korea through the following links. Please follow the two following links.
https://m.blog.naver.com/kma_131/222073284939
https://www.weather.go.kr/HELP/html/help_fct008.jsp
In the ‘Materials and Methods’ section (lines 118-129)
Generally, rainfall events can have different impacts depending on the amount of rainfall over a short period. Therefore, the degree of influence of rainfall intensity on PM wash-off may vary across the hourly rainfall magnitudes (i.e., short-term intensive rainfall) than the daily cumulative rainfall. According to the Korea Meteorological Administration data (KMA), rainfall intensity is generally classified into four categories: light (<3 mm/h), moderate (3–15 mm/h), heavy (15–30 mm/h), and violent (>30 mm/h), based on the rate of precipitation, which depends on the considered time. This study analyzed net PM removal ability change during natural rainfall events (19.5-mm heavy rain over three h) on surface PM and in-wax PM from the inner and outer crown-positioned leaves in five tree species. The Meteorological-related information referred to the KMA. Also, rainfall data were obtained from KMA Weather Data Service.
Point 9: L108-109: so you only considered heavy rainfall, if I understood correctly.
Response 9: Previous studies generally consider rainfall duration, pattern, and intensity as a function of rainfall characteristics to PM retention and wash-off dynamics (Xu et al. 2017; Xu et al. 2019; Zhang et al. 2019). These results suggest that PM retention and wash-off ability depended on rainfall intensity and plant species. Moreover, Xu et al. (2017) demonstrated that it is limited in real rainfall by its lower intensity and distribution within the tree canopy. So, based on these findings, we only considered heavy rainfall to understand the effects of rainfall in relation to leaf traits.
References
Xu, X., Zhang, Z., Bao, L., Mo, L., Yu, X., Fan, D., & Lun, X. (2017). Influence of rainfall duration and intensity on particulate matter removal from plant leaves. Science of the Total Environment, 609, 11-16.
Xu, X., Yu, X., Bao, L., & Desai, A. R. (2019). Size distribution of particulate matter in runoff from different leaf surfaces during controlled rainfall processes. Environmental Pollution, 255, 113234.
Zhang, L., Zhang, Z., Chen, L., & McNulty, S. (2019). An investigation on the leaf accumulation-removal efficiency of atmospheric particulate matter for five urban plant species under different rainfall regimes. Atmospheric Environment, 208, 123-132.
Point 10: L119-121: was this rainfall simulated or real? if real how did you measure that? and how you can discern between rainfall of 40mm and other heavier or lighter rainfall? this is not very clear to me
Response 10: Field experiments were conducted during natural rainfalls. It has already been described at the end of section '2.1'. The Meteorological-related information referred to the Korea Meteorological Administration data (KMA). Also, rainfall data were obtained from KMA Weather Data Service.
Point 11: L132-134 AND THE WHOLE TEXT: as mentioned before is more common to refer to these traits as morphological (macro) and anatomical (micro) traits
Response 11: Same as the response to Point 6 above. Please refer to Response 6.
Point 12: L158-159: so the anatomical parameters weren't calculated from the experimental samples?? why is that?
Response 12: In the field, the phenotypic plasticity of leaves is caused by various environmental factors in general, and these affect potential function. Therefore, we measured the leaf microstructure sampled from plants grown under controlled-environment conditions of the National Institute of Forest Science in order to observe a unique property on the leaves of each species.
Point 13: L213-216: this is not true, because for inner there is difference, I'm aware you added the reference to Figure1b which refer to outer but from the sentence it has another meaning, please specify. Figure 1: it is not very clear without reading the caption that graph a is for inner and b for outer leaves. You can write it on the graph. Figure 2 : I would suggest to add letters for significant difference as the graph before.
Response 13: As per your comments, we modified the sentence as follows. Also, we mentioned once again each description of the inner and outer crowns at the end of the sentence to emphasize the SPM10 in the 'Discussion' section.
In the ‘Results’ section (lines 233-239)
During the rainfall event, there were significant differences among tree species in particle size fractions for both crown-positioned leaves, except WPM10 (Figure 1a) and WPM2.5 (Figure 1b). Of note, SPM10 was an obvious difference among species in both inner and outer crown-positioned leaves (Figure 1). Inner crown-positioned leaves showed significant differences among species for SPM2.5, SPM10, and WPM2.5 during the rainfall event (Figure 1a); Outer crown-positioned leaves showed significant differences in SPM2.5, SPM10, and WPM10 (Figure 1b).
Point 14: Figure 1: it is not very clear without reading the caption that graph a is for inner and b for outer leaves. You can write it on the graph.
Response 14: (line 248) We added the letters INNER and OUTER to the graph of Figure 1.
Point 15: Figure 2 : I would suggest to add letters for significant difference as the graph before
Response 15: Figure 2 shows statistical comparisons between inner and outer crown-positioned leaves in particle size fractions using a T-test, not a comparison among species. Thus, we have already shown the statistical significance of differences (denoted by asterisks) in Figure 2. A significant difference between inner and outer crown-positioned leaves in each PM particle size fraction was indicated with asterisks by t-test (* p < 0.05; ** p < 0.01; *** p < 0.001; ns: not significant, p > 0.05).
Point 16: L462-465: Do you mean that PM which was wash-off and has sedimentation in the soil can be resuspended again and sedimented again on leaves? can you say more about it?
Response 16: We greatly appreciate Reviewer 2's comment that helped improve the manuscript. As per your comment, we revised the sentence as follows.
In the ‘Discussion’ section (lines 476-484)
Urban vegetation and building surfaces can improve air quality by blocking and retaining airborne PM pollution [14]. Some PM on the leaves can be encapsulated in the cuticle [28]. Previous studies [13,14] suggested that most PM particles on leaf surfaces are only temporarily retained on leaf surfaces before being resuspended in the air by winds or into the soil by rainfall. Ould-Dada and Baghini [28] suggested that resuspension of some particles can occur in all layers of the tree canopy. Particles deposited in tree canopy locations with low wind speeds are low are less susceptible to resuspension than particles located at the top of the canopy with high wind speeds. Wind speed can vary by tree height, thus being higher at the top and bottom of the tree canopy and lower in the middle.
Point 17: L475: was it true for all the species tested?
Response 17: We corrected the sentence by adding some trees with favorable efficiencies for PM adsorption.
In the ‘Discussion’ section (lines 502-506)
Trees that comprise a large portion of urban forests can play a vital role as the first line of defense to improve air quality by intercepting atmospheric noxious pollutants (i.e., harmful gases and particulates) by tree canopies such as Cassia fistula, Ficus benghalensis, Ficus elastica, Ficus rubra, Magnolia grandiflora, Pinus pinea, Platanus × acerifolia, and Tectona grandis [29,30].
Point 18: L495-499: please avoid repetition
Response 18: (line 515) As per your comment, it was deleted.
Point 19: L500: please format the reference in the right way
Response 19: (line 515) Deleted a citation because of a repeated sentence. Please see lines 513–515.
Point 20: L594-597: please avoid repeating "in this study". You can add "here"
Response 20: To avoid repetition, we deleted “in this study” in the sentence and revised it as follows.
In the ‘Discussion’ section (lines 628-631)
Despite differences in micro and macro morphological traits, five species in this study showed potential to ensure surface areas that can newly trap atmospheric particles due to the PM wash-off from leaf surfaces under high rainfall intensity [11,14].

Reviewer 3 Report
The research paper fits within the general scope of Horticulturae. The authors conducted a nice experiment the significance of the content is high as well as the interest of the readers of Horticulturae MDPI and the scientific soundness.
I have some minor comments and suggestions before accepting the paper for publication:
1) I urge the authors in the abstract section to report the differences between treatments in percentage in order to be clearer for the readers of Horticulturae.
2) It is not enough in the introduction section to report that few studies have been published before but rather what was the 'new' hypothesis of this current paper.
3) The discussion section should be absolutely improved it is not enough to report that your results are in line or not with previously published data but why and what is/are the main mechanisms behind these differences.
4) The conclusion section is weak and should be re-written.
based on the previous comments the paper can be accepted for publication after minor revisions.
Author Response
Author Reply to the Review Report
(Manuscript ID: horticulturae-2110117)
Dear Editor,
Title
Understanding particulate matter retention and wash-off during rainfall in relation to leaf traits of urban forest tree species
We are pleased to resubmit for publication the revised version of Manuscript ID: horticulturae-2110117 entitled “Understanding particulate matter retention and wash-off during rainfall in relation to leaf traits of urban forest tree species”. We are very grateful to the Associate Editor and three Reviewers for their deep and detailed comments which have helped us to improve our manuscript. We have revised our manuscript according to the suggestions and comments of three anonymous reviewers. Grammatical error correction and English improvement were carried done by a native-speaking English editor as suggested. Please find below point-by-point responses to each of the reviewers’ comments on our manuscript. Upload files of revised manuscripts were attached in two forms: ‘Track Changes’ and ‘Final PDF version’. Finally, we hope that reviewers and editors will be satisfied with the response to reviewer comments.
Sincerely yours,
Response to Reviewer 3 Comments
The research paper fits within the general scope of Horticulturae. The authors conducted a nice experiment the significance of the content is high as well as the interest of the readers of Horticulturae MDPI and the scientific soundness.
I have some minor comments and suggestions before accepting the paper for publication: based on the previous comments the paper can be accepted for publication after minor revisions.
Point 1: I urge the authors in the abstract section to report the differences between treatments in percentage in order to be clearer for the readers of Horticulturae.
Response 1: We greatly appreciate Reviewer 3's comment that helped improve the manuscript. As per your comments, we modified the ‘Abstract’ section as follows.
In the ‘Abstract’ section (lines 13-28)
Dynamic particulate matter (PM) behavior on leaves depends on rainfall events, leaf structural and physical properties, and individual tree crowns in urban forests. To address this dependency, we compared the observed relationships between PM wash-off ability and leaf traits on inner and outer crown-positioned leaves during rainfall events. Data showed significant differences in its PM wash-off ability between inner and outer crown-positioned leaves relative to rainfall events due to leaf macro- and micro-structure and geometric properties among tree species. Our results showed that PM wash-off effects on leaf surfaces were negatively associated with trichome density and size of leaf micro-scale during rainfall events. Specifically, Quercus acutissima with dense trichomes and micro-level surface roughness with narrow grooves on leaf surfaces showed lower total PM wash-off in both inner (-38%) and outer (105%) crowns during rainfall. Thus, their rough leaves in the inner crown might newly capture and/or retain more PM than smooth leaves even under rainfall conditions. More importantly, Euonymus japonicus, with a thin film-like wax coverage without trichome, led to higher total PM wash-off in both inner (368%) and outer (629%) crowns during rainfall. Furthermore, we studied the changes in PM wash-off during rainfall events by comparing particle size fractions, revealing a very significant association with macro-scale, micro-scale, and geometric features.
Point 2: It is not enough in the introduction section to report that few studies have been published before but rather what was the 'new' hypothesis of this current paper.
Response 2: As requested, we clearly indicated the specific hypotheses of the present study in the ‘Introduction’ section.
In the ‘Introduction’ section (lines 71-91)
In many contexts, field experiments are subject to many limitations due to other environmental interference factors such as atmospheric humidity, temperature, and wind speed; thus, the rainfall-induced wash-off processes have been proactively carried out using rainfall simulation experiments [7,17,21,22]. Most field experiments have been studied using the deposition models, such as PM and canopy interception modeling at tree canopy levels. Fewer studies on PM wash-off efficiency in the inner and outer crowns of individual trees due to rainfall have been studied [12]. Given the interest in direct field measurements, few studies have studied PM wash-off efficiency in individual trees' inner and outer crowns during rainfall intensity. On the other hand, extending the in-chamber seedling experiment to large trees growing in cities cannot be easy; hence, additional field research is required to explore the leaf potential in dynamic PM retention and wash-off processes in tree crowns [8,20]. It is necessary to recognize the barrier functions of urban forests to PM particles resuspended from urban lands containing different pollutants and to summarize information regarding the complex phenomenon of PM retention, wash-off, and resuspension under rainfall events. Here, we hypothesized that rainfall intensity could change PM retention and wash-off efficiencies depending on the inner and outer canopy location of urban tree species. These changes may depend on the unique microstructure and tree-specific crowns. Therefore, we described (1) the PM retention and wash-off in inner and outer crown-positioned leaves under natural rainfall intensity and (2) the relationship between PM retention and wash-off processes and leaf micro-structural factors.
Point 3: The discussion section should be absolutely improved it is not enough to report that your results are in line or not with previously published data but why and what is/are the main mechanisms behind these differences.
Response 3: We greatly appreciate Reviewer 3's comment that helped improve the manuscript. We revised the manuscript to demonstrate the validity and causation of key findings related to our research questions in the ‘Discussion’ section.
Point 4: The conclusion section is weak and should be re-written.
Response 4: As per your comments, we revised the ‘Conclusion’ sentence as follows.
In the ‘Conclusion’ section (lines 656-681)
Our field study uncovers the vital role of leaf macro-scale, micro-scale, and geometric features concerning structural descriptors for PM wash-off dynamics during rainfall events in leaf samples taken from the inner and outer tree crowns of tree species. During rainfall events, there were significant differences among tree species in particle size fractions for inner and outer crown-positioned leaves. In summary, rainfall intensity affects the PM wash-off in inner and outer tree crowns. In particular, the rainfall intensity is lower in the inner crowns than in the outer crowns. Inner crown-positioned leaves suppressed PM wash-off via cuticle hydration due to decreased rainfall intensity. In general, the wash-off levels of different-sized PM particles from leaves (especially PM10 deposited on leaf surfaces) on the ground increased after rainfall, except in Q. acutissima. Different tree species showed variable response potential for PM wash-off and/or retention during rainfall; E. japonicus and M. glyptostroboides without trichomes were considerably higher in PM wash-off levels on leaf surfaces. In contrast, Q. acutissima with micro-level surface roughness with dense and narrow grooves showed the lowest PM wash-off levels. More importantly, E. japonicus could play an essential role in removing PM pollution adsorbed on leaves during the descent of raindrops to the ground, cleaning leaf surfaces, and depositing the PM into the soil. Correlations between Several leaf surface traits and PM particle size fractions showed a vital transition between inner and outer crown-positioned leaves. Notably, in high rainfall intensity, PM wash-off increases with leaf turbulence, and this feature is highly dependent on macro- (W/L ratio and roughness index) and micro-structure (stomatal density) and geometric properties (wax and contact angle) of leaf surfaces. Our findings indicate slight correlations with microstructure in the PM wash-off of large particles during rainfall, whereas strong correlations were presented with small particles. Therefore, PM wash-off effects by individual tree crowns can be helpful for a comprehensive understanding of urban trees, atmospheric quality reduction, and urban hydrology in improved management and planning of urban forests and urban green areas.

Round 2
Reviewer 2 Report
Dear Authors I believe the manuscrip has very much improved after revision and it is now suitable for publication.